# Comparative analysis of viral RNA signatures on different RIG-I-like receptors

Raul Y Sanchez David[1,2], Chantal Combredet[1], Odile Sismeiro[3], Marie-Agnès Dillies[3], Bernd Jagla[3], Jean-Yves Coppée[3], Marie Mura[1,4], Mathilde Guerbois Galla[1†], Philippe Despres[5], Frédéric Tangy[1*], Anastassia V Komarova[1*]

[1]Unité de Génomique Virale et Vaccination, Institut Pasteur, CNRS UMR-3569, Paris, France; [2]Ecole doctorale, Biochimie, Biothérapies, Biologie Moléculaire et Infectiologie (B3MI), Université Paris Diderot - Paris 7, Paris, France; [3]Transcriptome and Epigenome, BioMics Pole, Center for Innovation and Technological Research, Institut Pasteur, Paris, France; [4]Unité Interactions Hôte-Agents Pathogens, Institut de Recherche Biomédicale des Armées, Brétigny-sur-Orge, France; [5]Technology Platform CYROI, University of Reunion Island, Saint-Clotilde, France

*For correspondence: ftangy@
pasteur.fr (FT); stasy@pasteur.fr
(AVK)

**Present address:** [†]University of
Texas Medical Branch,
Galveston, United States

**Competing interests:** The
authors declare that no
competing interests exist.

**Reviewing editor:** Xuetao Cao,
Zhejiang University School of
Medicine, China

**Abstract** The RIG-I-like receptors (RLRs) play a major role in sensing RNA virus infection to initiate and modulate antiviral immunity. They interact with particular viral RNAs, most of them being still unknown. To decipher the viral RNA signature on RLRs during viral infection, we tagged RLRs (RIG-I, MDA5, LGP2) and applied tagged protein affinity purification followed by next-generation sequencing (NGS) of associated RNA molecules. Two viruses with negative- and positive-sense RNA genome were used: measles (MV) and chikungunya (CHIKV). NGS analysis revealed that distinct regions of MV genome were specifically recognized by distinct RLRs: RIG-I recognized defective interfering genomes, whereas MDA5 and LGP2 specifically bound MV nucleoprotein-coding region. During CHIKV infection, RIG-I associated specifically to the 3' untranslated region of viral genome. This study provides the first comparative view of the viral RNA ligands for RIG-I, MDA5 and LGP2 in the presence of infection.

## Introduction

Because of increasing global population and worldwide exchanges, infections by RNA viruses are expanding with a huge impact on public health. New infections are emerging almost every year (pandemic flu, West Nile encephalitis, Severe Acute Respiratory Syndrome, Middle East Respiratory Syndrome, chikungunya, Ebola), tropical fevers are expanding (dengue, zika) and even previously controlled diseases are re-emerging (measles, poliomyelitis). Although several efficient vaccines against RNA viruses are used in human medicine, we still lack potent therapeutic treatments for the majority of viral infections as well as rational strategies to create efficient new vaccines. Recent progress in our understanding of cellular pathways controlling viral replication suggests that modulating host cell functions early upon viral infection could inhibit a large panel of RNA viruses (*Cheng et al., 2011*; *Shirey et al., 2011*; *Es-Saad et al., 2012*; *Guo et al., 2012*; *Lucas-Hourani et al., 2013*). The RIG-I-like receptors (RLRs) appear to be located at the frontline of the evolutionary race between viruses and the host immune system (*Vasseur et al., 2011*). These are cellular proteins that detect invasion of viral nucleic acid species inside the cytoplasm and initiate innate immune responses against viral infections that limit virus replication and trigger an adequate adaptive immune response

**eLife digest** An immune system can protect against disease-causing microbes such as viruses. Human cells contain three different receptors that can recognize and respond when a virus enters and begins to replicate inside. These receptors include RIG-I, MDA5 and LGP2, and they are collectively known as the RIG-I-like receptors. RIG-I-like receptors specifically recognize viruses that store their genetic material in the form of molecules of RNA. However, the specific viral parts that trigger RIG-I-like receptors to respond remain almost completely unknown.

RNA viruses include well-known and re-emerging viruses such as polio and measles, as well as chikungunya – a virus that is spread by mosquitoes and causes illness worldwide. This means that understanding how RIG-I-like receptors identify RNA viruses and then trigger an immune response to eradicate them has the potential to inform the development of vaccines and antiviral therapies for many diseases.

Sanchez David et al. now describe and validate a new experimental approach to determine the distinct viral regions that are recognized by human RIG-I-like receptors. The approach involves purifying the RIG-I-like receptors out of infected cells and then working out the sequence of RNA fragments that bind to the receptors. This approach revealed that each of the three human RIG-I-like receptors detected different viral RNA sequences during a measles infection. On the other hand, only RIG-I could recognize a specific part of the chikungunya virus genome.

All together, the experiments illustrate how to identify the RNA sequences recognized by any of the three human RIG-I-like receptors during infection by a RNA virus. With the ability to gain this kind of insight, it may soon be possible to develop ways of using the RIG-I-like receptor pathway to control viral infections and enhance the body's immune response to vaccination.

(*Errett and Gale, 2015*). RLR signalling operates ubiquitously. Hence, RLRs represent attractive strategies for antiviral therapy and vaccine development.

The RLR family of pattern recognition receptors (PRRs) is a group of cytosolic RNA helicases that can identify viral RNA as non-self *via* binding to pathogen-associated molecular pattern (PAMP) motifs. To date three RLR members have been identified: RIG-I (Retinoic acid-Inducible Gene-I), MDA5 (Melanoma-Differentiation-Associated gene 5), and LGP2 (Laboratory of Genetics and Physiology 2) (reviewed in [*Loo and Gale, 2011*; *Dixit and Kagan, 2013*]). They share a number of structural similarities including their organization into three distinct domains (*Figure 1A*): i) an N-terminal region consisting of tandem caspase activation and recruitment domains (CARD), ii) a central DExD/H box RNA helicase domain with the capacity to hydrolyze ATP and to bind RNA, and iii) a repressor domain (RD) embedded within the carboxy-terminal domain (CTD). These RNA helicases interact with particular signatures of viral RNA, most of which are still unknown. Upon ligand recognition, RLRs bearing the CARD domain (MDA5 and RIG-I), undergo a conformational change that permits the CARD domain to be recruited and to oligomerize with MAVS either in the peroxisome or the mitochondrion. This activates signalling pathways leading to translocation of the transcription factors IRF3, IRF7 and NF-kB into the nucleus to initiate expression and secretion of type I IFNs and other proinflamatory cytokines. Secreted type I IFNs bind to their receptors and activate the JAK/STAT signalling pathway inducing the expression of more than 300 IFN-stimulated genes (ISGs) bearing IFN-stimulated response elements (ISREs) (*de Veer et al., 2001*). If the virus has no means for subverting the interferon pathway, the infected tissue turns into an antiviral state leading to i) apoptosis of the infected cells, ii) limited propagation of the virus by the expression of ISGs in neighbouring cells and iii) generation of a cytokine storm that triggers the specific adaptive immune response as well as favouring immune cell infiltration from the cardiovascular system.

Members of the RLR family have been implicated in the recognition of a variety of viruses (*Dixit and Kagan, 2013*; *Goubau et al., 2013*; *Patel and Garcia-Sastre, 2014*). RIG-I confers recognition of hepaciviruses and of members of the *Paramyxoviridae, Rhabdoviridae*, and *Orthomyxoviridae* families. For example, 5'copy-back defective-interfering (DI) genomes produced by numerous negative-sense RNA viruses specifically associate with RIG-I and activate IFN induction (*Strahle et al., 2006*; *Baum et al., 2010*; *Komarova et al., 2013*; *Runge et al., 2014*). MDA5

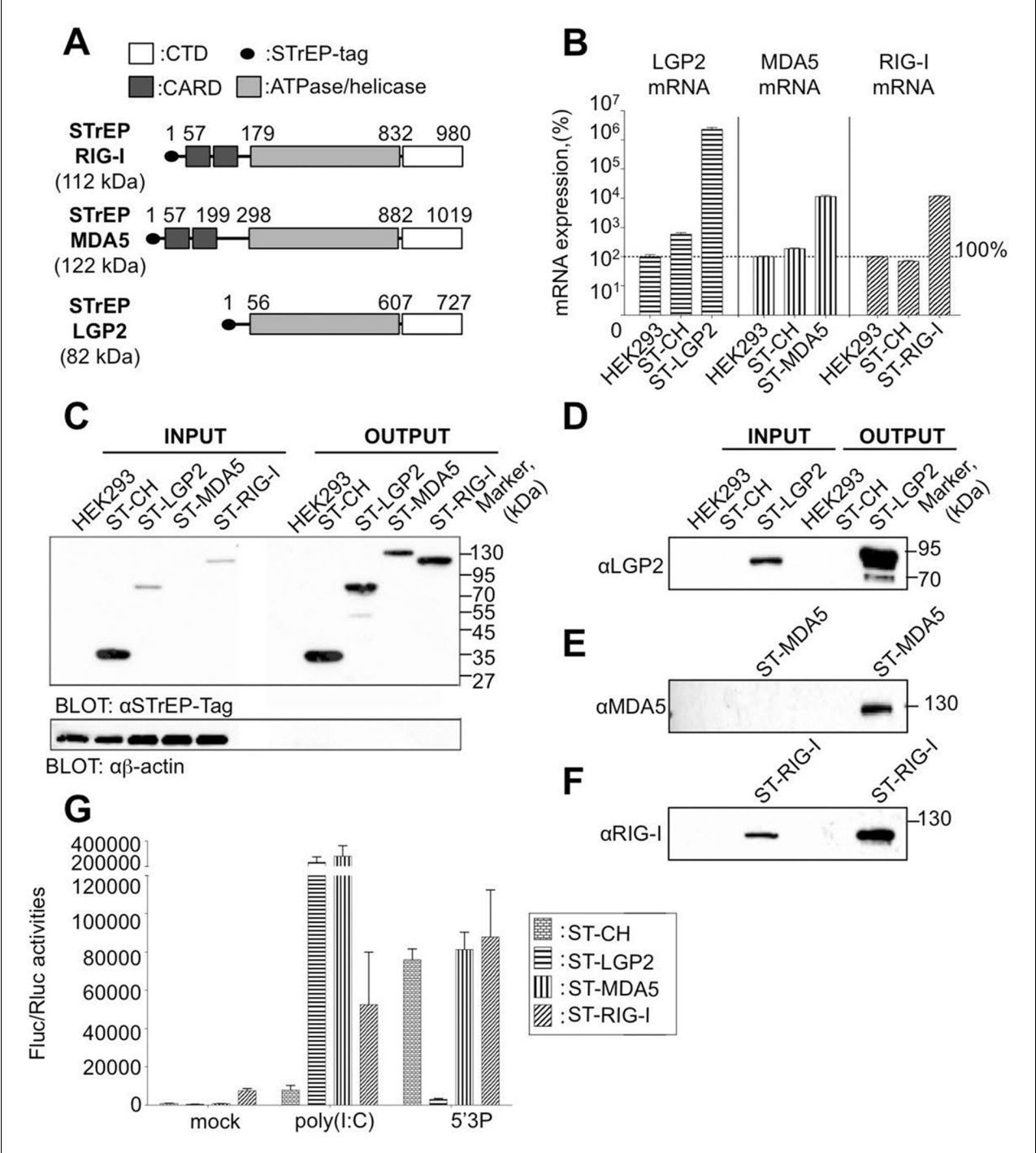

**Figure 1.** Rig-I Like Receptor (RLR) gene expression in stable cell lines encompassing ST-RLRs. (**A**) Schematic representation of the protein domains for each RLR. Domain boundaries are indicated for human RIG-I, MDA5, and LGP2 proteins according to interpro (www.ebi.ac.uk/interpro). (**B**) LGP2, MDA5 and RIG-I mRNA levels in ST-RLR cells. RLR mRNA expression were calculated by relative RT-qPCR using specific probes for LGP2, MDA5 or RIG-I (on 100 ng of total RNA). Ct were normalized using a specific probe against GAPDH house keeping gene. Percentage of mRNA expression was done by setting HEK293 cells as 100% of gene expression for each probe. Samples were analyzed in triplicates with standard deviation represented on the figure. (**C–F**) Analysis of RLR protein expression in ST-RLR cells and efficiency of tagged RLR purification by affinity chromatography. ST-RLR cell lysates (INPUT) were affinity-purified using STrEP-Tactin beads (OUTPUT). Western blot analysis was performed using (**C**) α-STrEP-Tag, (**D**) anti-LGP2, (**E**) anti-MDA5 or (**F**) anti-RIG-I antibodies. (**G**) IFNβ promoter activity assay in ST-RLR cells. Cells were transfected with pIFNβ-FLuc, pTK-Rluc and either mock, poly(I:C) or 5'3P.

detects members of the *Picornaviridae*, *Caliciviridae*, and *Coronaviridae*. *Flaviviridae*, *Reoviridae*, and *Arenaviridae* are detected by both MDA5 and RIG-I. LGP2 can act both positively or negatively upon activation by different viruses (*Moresco and Beutler, 2010*; *Deddouche et al., 2014*). Several DNA viruses have also been reported to activate the RLR pathway, including Herpes simplex virus-1, Adenovirus, Epstein-Barr virus, Vaccinia virus and Hepatitis B virus (*Choi et al., 2009*; *Sato et al., 2014*). In the case of DNA viruses, poly dA:dT DNA sequences trigger IFN responses after RNA polymerase III transcription and detection by RIG-I (*Ablasser et al., 2009*; *Chiu et al., 2009*). Surprisingly, intracellular bacteria, *Listeria monocytogenes*, *Legionella pneumophila*, *Salmonella typhimurium*, *Shigella flexneri* also activate type I IFN responses through RIG-I signalling and *Plasmodium* RNAs are sensed by MDA5 during malaria infection (*Chiu et al., 2009*; *Monroe et al., 2009*; *Liehl et al., 2013*; *Patel and Garcia-Sastre, 2014*).

Although human RLRs have recently received considerable attention, to our knowledge, nobody has yet simultaneously explored viral RNA partners that bind the three known RLRs under the same experimental conditions. In addition, only few studies characterised molecular features of the RLR ligands in the presence of viral infection (*Baum et al., 2010*; *Deddouche et al., 2014*; *Runge et al., 2014*). Thus, it is difficult from existing observations to get a clear picture of i) the biological ligands for each of the RLRs and ii) the functional differences between RLRs.

To study virus-host RNA-protein interactions during viral infection, we previously developed and validated a high-throughput riboproteomic approach based on One-STrEP-tagged protein affinity purification and next-generation sequencing (NGS) (*Komarova et al., 2013*). This protocol allows exploring biologically active macromolecular complexes within infected cells. Here, we applied this method to study viral RNA signatures sensed by RIG-I, MDA5 and LGP2 cytosolic receptors upon infection with both negative- and positive-sense RNA viruses. Such study provides a better understanding of the mechanisms and roles of RLR associated RNAs. It helps to conceive new therapeutic approaches such as modulators of innate immunity or new vaccine adjuvants.

## Results

### Generation of stable cell lines expressing One-STrEP-tagged RIG-I, MDA5 and LGP2

To identify and compare RLRs viral RNA partners upon infection with different viruses, we generated human HEK293 cell lines stably expressing One-STrEP-tagged LGP2, or MDA5, or RIG-I proteins (*Figure 1A*). One-STrEP tag provides a reliable, rapid and efficient protocol for ribonucleoprotein (RNP) and protein complexes purification (*Komarova et al., 2011*; *2013*). We tagged the RLRs N-terminally since previous studies have shown that adding N-terminal tags to RLRs had no negative effect on functional interactions with their RNA ligands (*Pichlmair et al., 2009*; *Rehwinkel et al., 2010*; *Zhang et al., 2013a*; *Deddouche et al., 2014*). These cell lines were assigned ST-RLR cells (ST-LGP2, ST-MDA5 and ST-RIG-I, respectively). In addition, a stable cell line (assigned ST-CH) expressing the Cherry protein instead of tagged RLRs was generated as a negative control to allow subtraction of non-specific RNA binding. Using RT-qPCR analysis and affinity purification followed by western blot analysis we validated the expression of tagged RLRs by each stable cell line (*Figure 1B–F*). We then analysed the functional profiles of the ST-RLR cell lines upon transfection with RIG-I and MDA5 agonists by using a classical IFN-β promoter activity assay (*Figure 1G*). We observed that transfection of ST-MDA5 and ST-LGP2 cells with MDA5 agonist poly(I:C) elicited an increased IFN-I response. In contrast, transfection of ST-LGP2 cells with the RIG-I agonist 5'triphosphate-RNA (5'3P) masked the IFN-I response (*Figure 1G*). These observations are in accordance with previous studies showing that over-expression of MDA5 or simultaneous co-expression of MDA5 and LGP2 synergize type I IFN response upon activation with MDA-specific agonists or upon infection with encephalomyocarditis virus (EMCV) (*Bruns et al., 2013*). LGP2 over-expression has also been previously found to inhibit RIG-I signalling (*Moresco and Beutler, 2010*). Altogether this demonstrates that the three stable cell-lines that we established express tagged RLRs that are functional and can be purified by affinity chromatography.

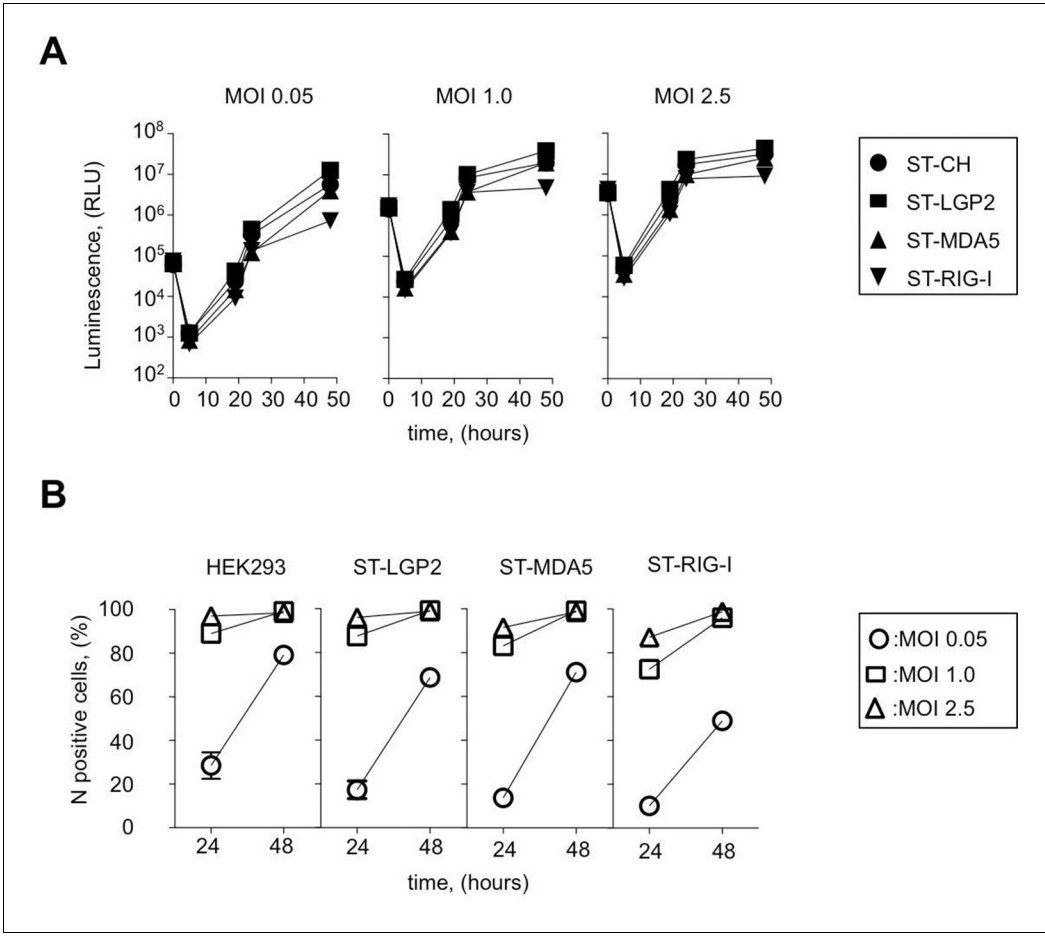

**Figure 2.** Efficiency of ST-RLR cells infection by negative-sense RNA virus (MV). (**A**) Efficiency of Fluc (rMV2/Fluc) expressing MV replication in ST-RLR cells. ST-RLR cells were infected with the rMV2/Fluc (MOIs: 0.05, 1, 2.5). Luc activity was analyzed 5, 19, 24 and 48 hr post-infection. (**B**) Efficiency of MV replication in ST-RLR analyzed by FACS. ST-RLR cells were infected by MV. After 24 and 48 hr, cells were harvested, fixed and stained using an anti-N antibody to measure percentage of N positive cells. Experiments were performed two times and data represent means ± SD of the technical triplicates of the most representative experiment.

## RNA virus replication in cells expressing tagged RLRs

To test whether the over-expression of RLRs influences the efficiency of RNA virus replication we infected ST-RLR stable cell lines with a representative of negative- or positive-sense RNA virus: measles virus (MV) and chikungunya virus (CHIKV), respectively.

MV belongs to the family *Paramyxoviridae* (order *Mononegavirales*) and is often considered as a prototypical member of negative-sense RNA viruses. MV genome is used as a template by the viral polymerase to replicate viral genome and to synthesize mRNA molecules encoding six structural proteins: N, P, M, F, H, L and two non-structural virulence factors: V and C. To determine the kinetics of MV replication in ST-RLR cells we used recombinant MV expressing the *Firefly* luciferase (*Fluc*) gene from the viral genome (rMV2-Fluc [*Komarova et al., 2011*]). We observed that at low multiplicity of infection (MOI) MV replication was slightly less efficient in ST-RIG-I cells than in the two other cell lines ST-MDA5 and ST-LGP2 (*Figure 2A*). MV replication was also monitored by immunostaining of MV nucleoprotein (MV-N) and flow cytometry analysis at 24 and 48 hr post-infection (*Figure 2B*). Again, we observed that viral replication was slightly reduced in cells expressing an additional copy of RIG-I protein, particularly at low MOIs.

CHIKV is a member of the family *Togaviridae* with a positive-sense single-stranded RNA genome. Non-structural proteins (NSP1-4) are directly translated from the viral genome (49S) at early stage of viral replication, whereas structural proteins C-E3-E2-6K-E1 (capsid C, envelope glycoproteins E3, E2

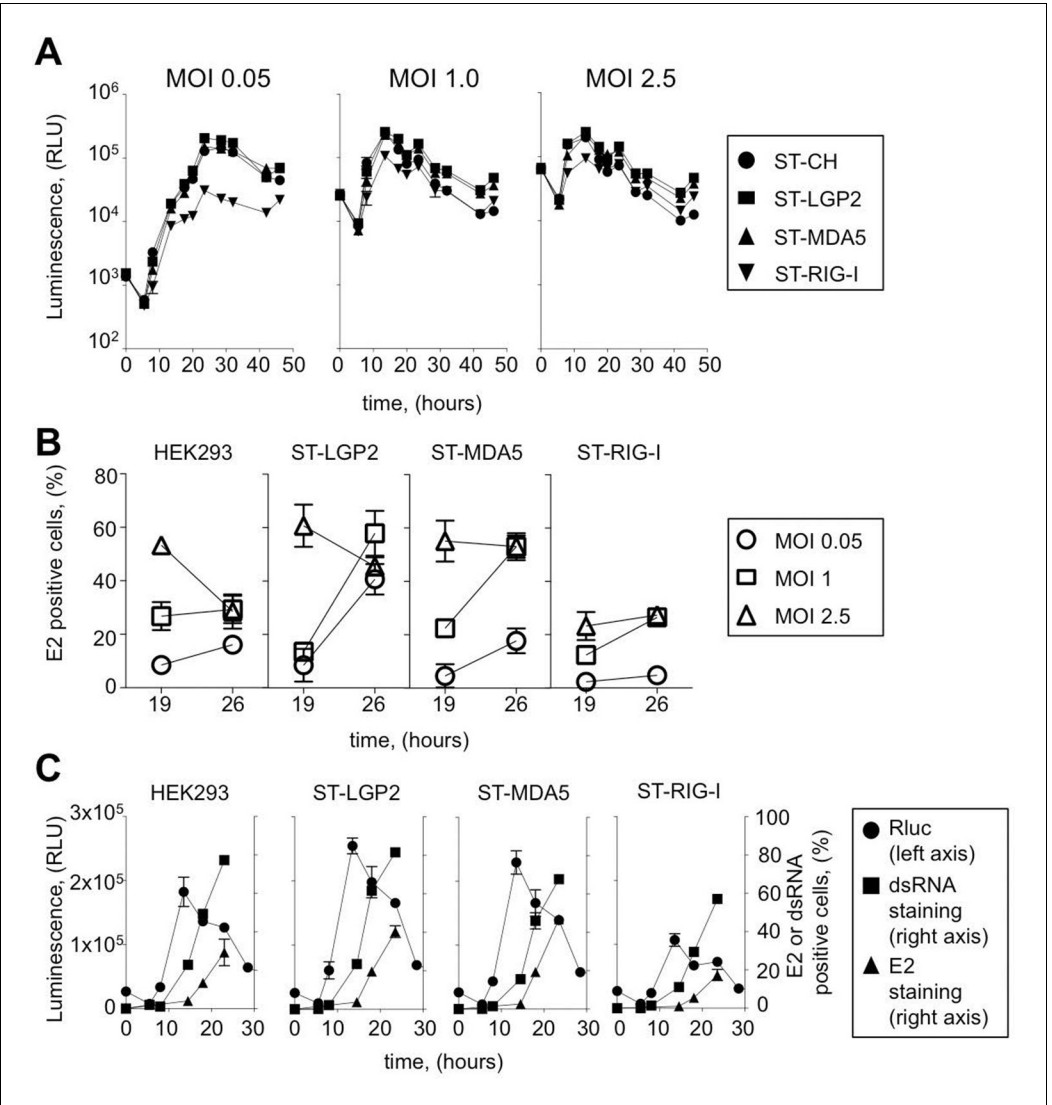

**Figure 3.** Efficiency of ST-RLR cells infection by positive-sense RNA viruses (CHIKV). (**A**) Replication efficiency of CHIKV-Rluc in ST-RLR cells. ST-RLR cells were infected with a CHIKV-Rluc (MOIs: 0.05, 1, 2.5). renLuc activity was analyzed 0, 5, 8, 10, 13, 19, 24, 32 and 40 hr post-infection. (**B**) Efficiency of CHIKV replication in ST-RLR cells analysed by FACS. ST-RLR cells were infected with *wt* CHIKV (MOIs: 0.05, 1, 2.5). Immunostaining of the E2 glycoprotein was performed and percentage of positive cells was determined. (**C**) Analysis of early and late steps of CHIKV replication. ST-RLR cells were infected with CHIKV-Rluc or *wt* CHIKV at an MOI 1. Rluc activity was measured for the CHIKV-Rluc infection (left axis) 6, 9, 15, 18, 23 and 28 hr post-infection. *wt* CHIKV-infected cells were harvested 6, 9, 15, 18 and 23 hr post-infection, stained with an antibody recognizing double stranded RNA or E2 (right axis) and percentage of positive stained cells was determined by FACS. Experiments were performed two times and data represent means ± SD of the technical triplicates of the most representative experiment.

The following figure supplement is available for figure 3:

**Figure supplement 1.** Experimental approaches used to determine early and late steps of CHIKV replication.

and E1 and the ion channel protein 6K) are encoded by subgenomic mRNA (26S mRNA) during the late viral cycle. To assess CHIKV replication kinetics in tagged RLR cell lines we used a molecular clone of CHIKV (CHIKV-Rluc) at different MOI: 0.05, 1, 2.5. In this virus the reporter *Renilla* luciferase (*Rluc*) gene is expressed as an integrated part of the non-structural polyprotein of CHIKV. The

CHIKV-Rluc allows monitoring the early stage of viral replication (*Henrik Gad et al., 2012*). CHIKV-Rluc replicated in the 4 cell lines as detected by strong Rluc expression and classical growth kinetics for this virus (*Figure 3A*). However, similarly to MV we observed that CHIKV replication was reduced at low MOI in ST-RIG-I cells. This observation was confirmed by measuring the accumulation of E2 glycoprotein, a late viral product, in ST-RLR cells during infection with *wt* CHIKV. Again, viral replication was delayed in cells expressing an additional copy of RIG-I protein (*Figure 3B*).

These experiments demonstrate that both negative- and positive-sense RNA viruses replicate in cells expressing additional copies of tagged RLRs. Thus, ST-RLR cells provide suitable conditions to study RNA ligands of RLRs within infected cells.

## Choice of the best time point to purify RLR-associated viral RNA ligands

As showed on *Figure 2B* all three ST-RLRs were efficiently infected with MV (MOI 1) at 24 hr post-infection (with more than 60% of MV-N positive cells). We have previously observed extremely valuable virus-host interaction networks in the same conditions of infection (*Komarova et al., 2011*). Therefore, we choose these conditions to purify RLR proteins complexed with MV RNA ligands.

In the case of CHIKV infection, we distinguished early and late stages of virus replication. Indeed, it is believed that at early stage of the viral cycle dsRNA intermediary products are sensed by RLR receptors to activate type I IFN signalling. CHIKV non-structural proteins translation and viral genome replication are performed at early steps of replication. This stage can be monitored using CHIKV-Rluc. We observed Rluc accumulation starting from 8 hr post-infection in the ST-RLR and in ST-CH cells (*Figure 3A and C* left axis). In addition, dsRNA presence was assessed in cells infected with *wt* CHIKV by using intracellular staining with an anti-dsRNA antibody and flow cytometry analysis. We observed dsRNA accumulation starting from 10 hr post-infection (*Figure 3C* right axis). Further, efficient translation of the structural E2 glycoprotein that corresponded to the late stage of CHIKV replication was observed at 15 hr post-infection in all three ST-RLR cell lines (*Figure 3B and C* right axis). Schematic representation of the three technical approaches is represented (*Figure 3— figure supplement 1*). These experiments showed that purification of CHIKV RNA ligands complexed with RLRs should be performed between 10 and 15 hr post-infection with CHIKV at MOI of 1 (*Figure 3C*).

## RNAs purified from tagged RLRs upon infection with MV and CHIKV are immunoactive

To purify the RNAs associated with RLRs upon infection with negative- or positive-sense RNA viruses, ST-RLRs cells were infected with MV or with CHIKV at MOI1. Cells were harvested 24 hr (MV) and 13 hr (CHIKV) post-infection and RLRs together with associated partners were purified from total cell lysates by affinity chromatography (see Materials and methods). The RNA molecules interacting with RLRs or the negative control CH were then extracted and designated as RLR/RNA or the negative control CH/RNA.

We tested the capacity of these RNA molecules to induce an IFN-mediated antiviral response. RNA samples extracted from LGP2, MDA5, RIG-I or CH were transfected into STING-37 reporter cells that stably express the *Fluc* gene under the control of a promoter sequence containing five IFN-Stimulated Response Elements (ISRE) (*Lucas-Hourani et al., 2013*). We observed that RNA ligands purified from LGP2 and RIG-I receptors in the presence of MV or CHIKV infections had an increased immunostimulatory activity in comparison with MDA5-specific RNA ligands (*Figure 4A and B*). As expected, RNAs purified from negative control CH provided poor immunostimulatory activity in the same experimental setup as well as RNA purified from RLR complexes in the absence of a viral infection (*Figure 4*). These results demonstrate that RNA ligands bound to ST-RLRs are specifically enriched for immunoactive RNA molecules, and that our experimental strategy is sensitive enough for the isolation of RLR-specific RNA ligands for both negative- and positive-sense RNA viruses within infected cells.

## RLR-dependent innate immune sensing of MV and CHIKV infections

To further evaluate the requirement of LGP2, MDA5 and RIG-I for IFN response to MV and CHIKV, we depleted STING-37 cells of LGP2, or MDA5, or RIG-I *via* transduction with lentiviral vectors

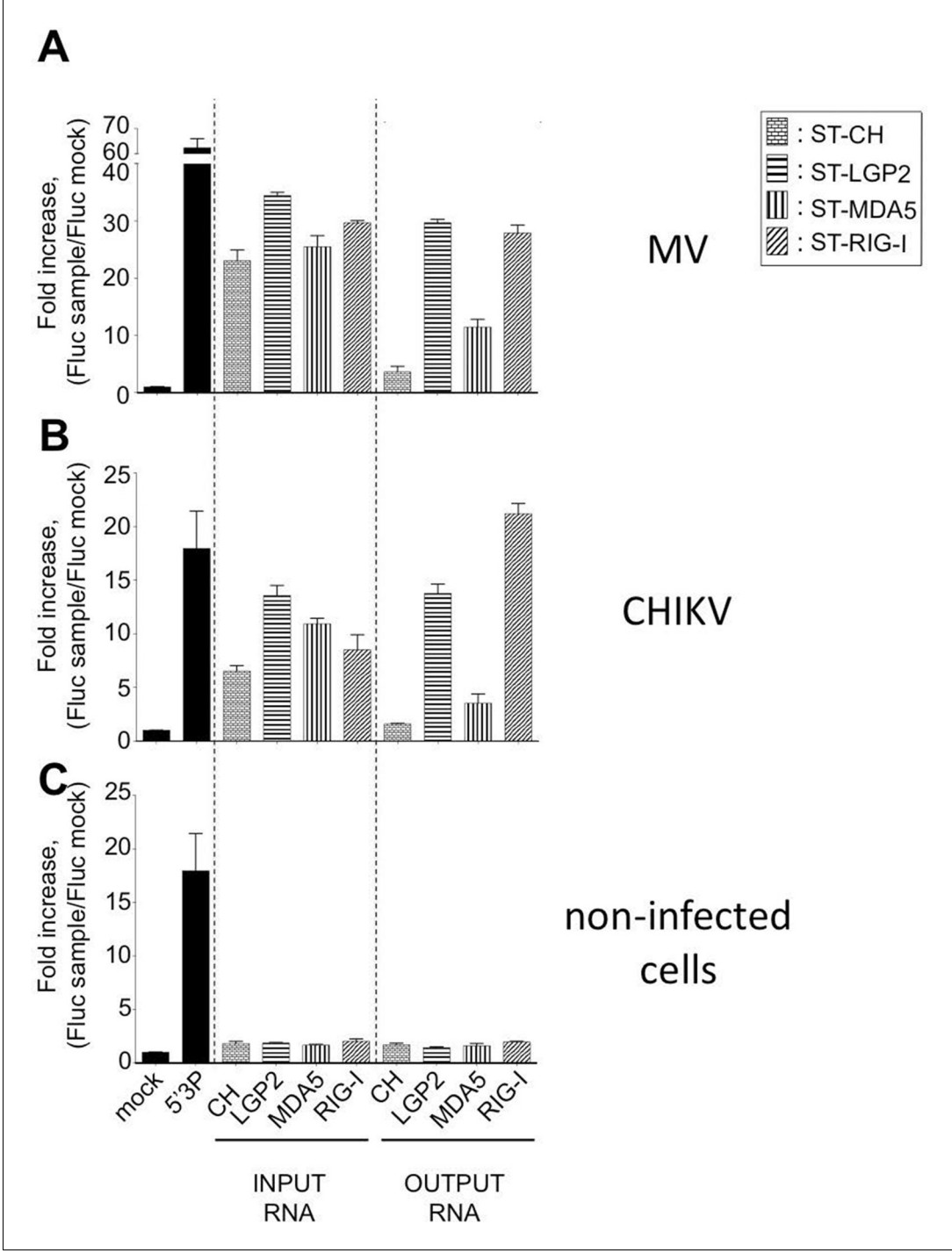

**Figure 4.** Immunostimulatory activity of RNA-ligands co-purified with ST-RLRs upon infection with MV and CHIKV. ST-RLR cells were infected with MV for 24 hr (**A**), CHIKV for 13 hr (**B**) or mock infected (**C**). Total cell lysate was used for total RNA purification (INPUT) and for affinity purification of RLR RNA complexes, followed by RNA extraction (OUTPUT). Immunostimulatory activity was assessed by transfection into STING-37 reporter cell lines (*Lucas-Hourani et al., 2013*). Fluc activity was measured and normalised to mock transfected cells, 5'3P was used as a positive control. Experiments were performed two times and data represent means ± SD of the technical triplicates of the most representative experiment.

expressing shRNA against each of these receptors or with a control shRNA. The level of LGP2, MDA5, or RIG-I mRNAs in puromycin-resistant transduced cell populations was assessed by quantitative RT-qPCR. Silencing decreased LGP2, MDA5 and RIG-I levels to 41%, 48% and 32%, respectively (*Figure 5A*). These cells were assigned STINGshLGP2, STINGshMDA5, STINGshRIG-I and STINGshNeg, respectively (see Materials and methods).

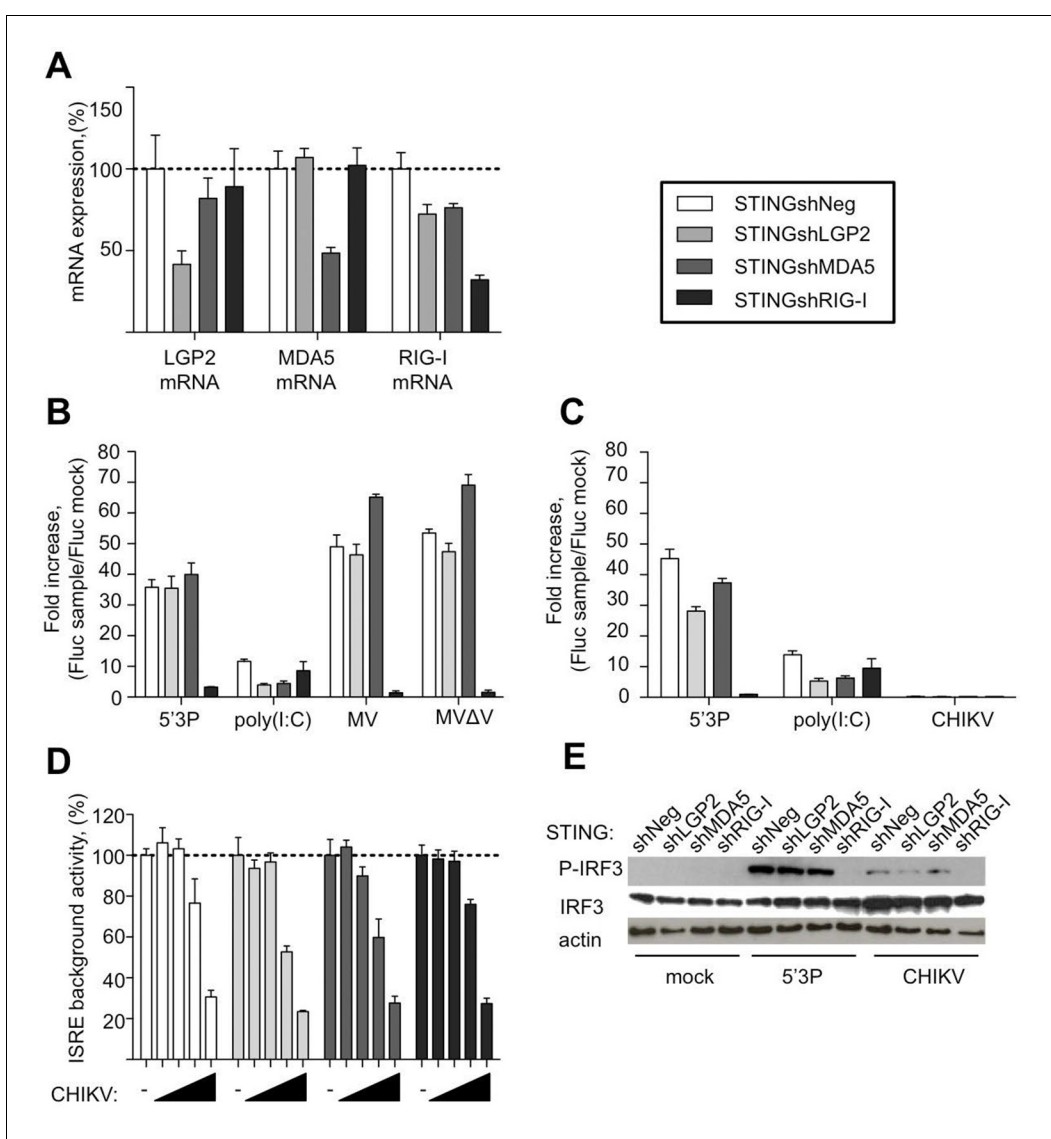

**Figure 5.** Innate sensing of MV and CHIKV infections by different RLRs. (A) LGP2, MDA5 and RIG-I mRNA levels in STINGshRLR cells. STING-37 reporter cell line was transduced by lentiviral vectors expressing an shRNA directed against either LGP2 (shLGP2),or MDA5 (shMDA5), or RIG-I (shRIG-I), or non-silencing (shNeg). qPCR analyses with specific probes against the mRNA of each of the RLRs were performed. Relative mRNA expression was done using GAPDH as reference gene and shNeg as reference sample. ISRE activation in STINGshRLR cells by MV or MVΔV (B) or CHIKV (C) infection. 5'3P and poly(I:C) were used as controls. Results are represented as fold increase of ISRE expression compared to mock infected cells. (D) Analysis of STINGshRLR promoter activity in the presence of different loads of CHIKV. MOIs 0; 0.001; 0.01; 0.1 and 1 were used. Fluc activity was measured 13 hr post-infection and normalized by setting MOI 0 as 100% of ISRE background activity for the corresponding cell line. (E) RLR-dependent IRF3 phosphorylation upon CHIKV infection. IRF3 phosphorylation was analysed in STINGshRLR cells after CHIKV infection by Western Blot using a specific antibody recognizing the phosphorylated form of IRF3. Experiments were performed three times and data represent means ± SD of the technical triplicates of the most representative experiment.

Upon MV infection only the cells that had RIG-I receptor silenced were strongly impaired in ISRE promoter activation, while MDA5 and LGP2 silencing did not affect signalling (*Figure 5B*). This was independent of the MV-V protein, which is known to subvert the RLR response through interaction with MDA5 and LGP2 (*Andrejeva et al., 2004*; *Childs et al., 2013*; *Motz et al., 2013*), as MVΔV virus deleted of the V protein behave similarly.

Upon CHIKV infection, we failed to detect any activation of ISRE promoter in the conditions previously established at 13 hr post-infection (*Figure 5C*). Further analysis showed that increasing CHIKV MOI reduced *Fluc* expression under the control of ISRE promoter in all silenced cell lines, even at early time post-infection (*Figure 5D*). Old-world alphaviruses are known to shutdown transcription in infected cells (*Garmashova et al., 2006*; *White et al., 2011*; *Akhrymuk et al., 2012*; *Bouraï et al., 2012*). However, CHIKV infection induces IRF3 phosphorylation and nuclear translocation (*White et al., 2011*). Therefore, we used western blot analysis of RLR-dependent type I IFN signalling to evaluate IRF3 phosphorylation in CHIKV-infected STINGshRLR cells. Interestingly, we observed that only in cells deficient in RIG-I, IRF3 was no longer phosphorylated (*Figure 5E*).

These data validated the important role of RIG-I in viral sensing for both MV and CHIKV infections. Using this experimental set up we failed to validate LGP2 and MDA5 implication in type I IFN activation. These results contradict the fact that immunoactive RNA ligands were co-purified with LGP2 from either MV- or CHIKV-infected cells and MDA5/RNAs possessed only slight immunostimulatory activity (*Figure 4A,B*). To clarify this question we proceeded to the NGS of RLR-associated RNAs.

## NGS analysis reveals that distinct regions of MV genome are specifically recognized by distinct RLRs

To identify the nature of RNA molecules bound to RLRs in an unbiased manner, strand-specific NGS analysis was performed using the Illumina technology. To distinguish between RLR-associated RNA molecules and non-specific binding to the beads, co-affinity purification experiments were performed in parallel on ST-CH cells. All experiments were performed in triplicates. Approximately, 16 million reads of 51 nucleotides in length were obtained from each sequencing sample and were mapped to the viral genome. This gave MV genome coverage ranging from 274X to 2433X. We also performed NGS analysis of total RNA samples. Total RNA results provided a classical example of the transcription gradient, which is the characteristic of the order *Mononegavirales*. Indeed, MV transcription is initiated from a single promoter at the 3' end of the genome and at each gene junction the virus RNA-dependent RNA-polymerase either falloff or continue the transcription of the downstream gene (*Cattaneo et al., 1987*; *Plumet et al., 2005*). Due to this gradient of transcription, MV promoter-proximal genes are expressed more efficiently than promoter-distal genes. The normalized read counts (see bellow) on the MV genome are shown in *Figure 6—figure supplement 1*, where only the first position of the reads is taken into account.

We performed statistical analysis of reads for each RLR that were enriched in the RLR/RNA samples compared to the CH/RNA samples, but lacked enrichment in total RNA of the RLR sample test compared to total RNA of the CH control sample. For this we performed differential analyses between RLR/RNA and CH/RNA on one hand, and between total RLR RNA and total CH RNA on the other hand. The normalization and differential analyses were performed with R (*Team, 2013*) and the DESeq2 package. A p-value adjustment was performed according to the Benjamini and Hochberg (BH) procedure (*Benjamini and Hochberg, 1995*). Positions were considered significantly enriched when their adjusted p-value was lower than 0.05. Importantly, there were no differences except for the ST-RIG-I cells for the total RNA profiles of the ST-RLR cells upon infection with MV (*Supplementary file 1*). These differences were taken into consideration upon statistical analysis. The distributions of normalized read counts matching the MV genome sequences were represented along the viral genome with the X axis corresponding to all possible positions on the viral genome, and the Y axis showing the normalized number of reads that begin at that position (*Figure 6A–C*).

The distribution of the normalized sequencing reads in the RLR samples collected during MV infection differed a lot for each of the three cytosolic receptors. Indeed, NGS data analysis showed specific MDA5 association with RNA sequences of positive polarity, which most likely correspond to the MV-N mRNA or a read-through transcript of the MV leader region containing N coding sequence (*Figure 6A*). LGP2 associated RNA sequences represented only a short part of the N coding region and were also enriched in L gene-derived RNAs with both positive and negative polarities (*Figure 6B*). Surprisingly, we failed to detect any RIG-I specific reads aligning to the MV genome

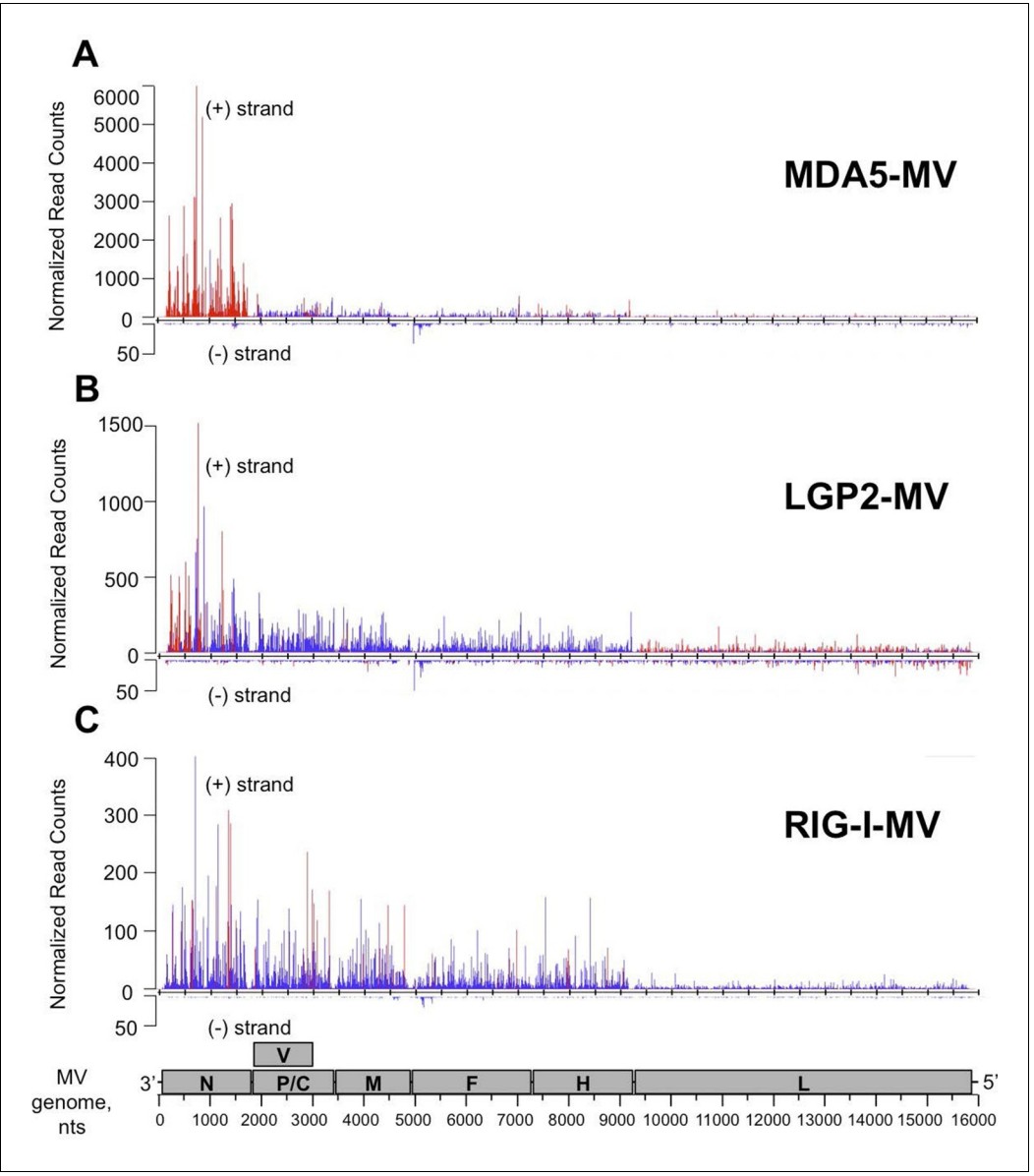

**Figure 6.** NGS analysis of specific RLR viral partners purified upon infection with MV. MDA5/RNA (**A**), LGP2/RNA (**B**) and RIG-I/RNA (**C**) samples were subjected to Illumina strand-specific NGS analysis. Sequencing reads were mapped to the MV genome and only the first nucleotide was retained. Differential analyses were performed between RLR/RNA and CH/RNA on one hand and total RLR/RNA and total CH/RNA on the other hand. The distributions of normalized read counts matching the MV genome are represented along the viral genome with the X axis corresponding to all possible positions on the MV genome, and the Y axis showing the normalized number of reads that begin at that position on the positive (+) or the negative (-) strand of the genome. Significantly enriched reads are represented in red and non-significantly enriched reads are in blue. N, P/V/C, M, F, H, L are MV genome regions coding for the corresponding proteins.

The following figure supplement is available for figure 6:

**Figure supplement 1.** NGS profile of total RNA aligned on the MV genome from ST-CH cells infected with the MV.

(not shown). This was unexpected since in the above-presented silencing experiment, RIG-I was the only important of the three RLRs sensor for innate immunity activation upon MV infection (*Figure 5B*). Thus, we performed a second RIG-I purification and NGS experiment. For this second RIG-I/RNA sequencing the viral genome coverage ranged from 144X to 1856X. Again, we obtain similar read profile with only few short positions along MV genome statistically enriched in the RIG-I/RNA sample (*Figure 6C*).

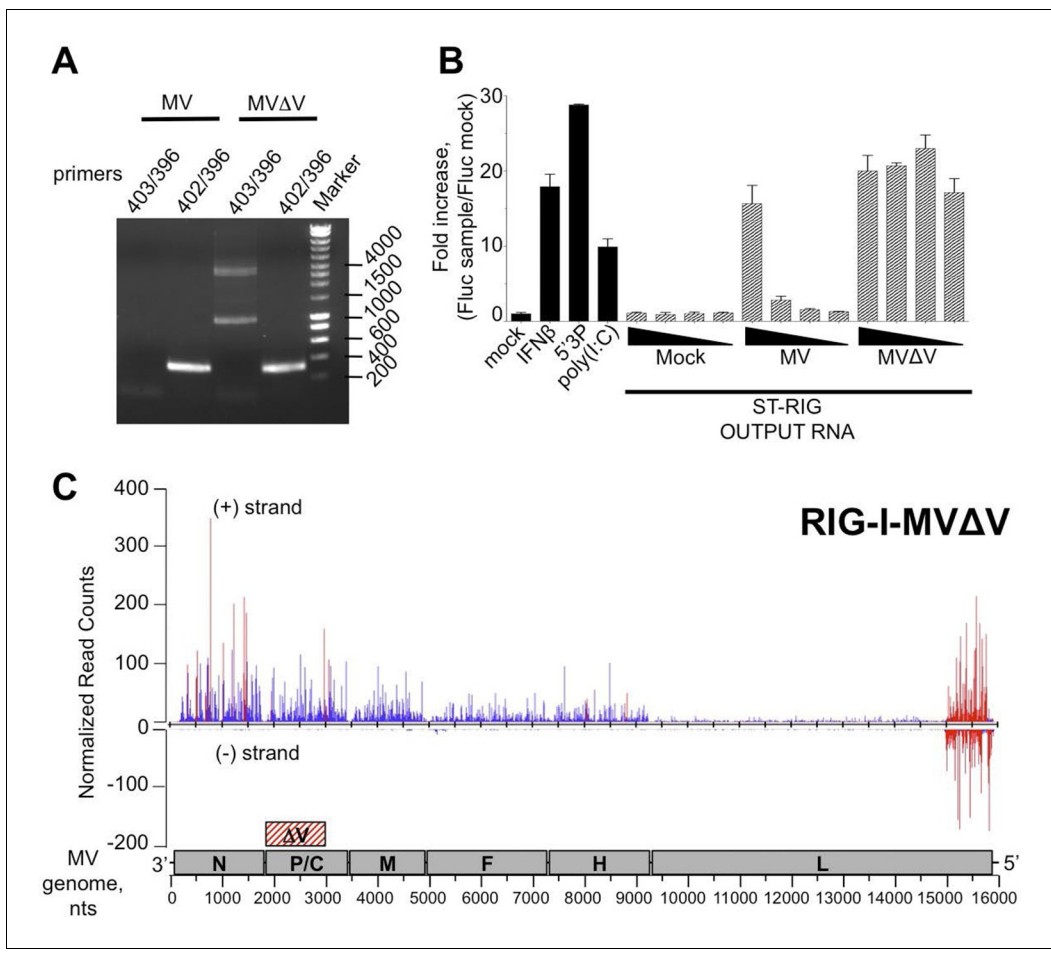

**Figure 7.** 5' copy-back DI-genome is specifically associated with RIG-I upon infection with MVΔV. (**A**) RT-PCR amplification of 5' copy-back DI-genome from cells infected with MVΔV using specific primers. Primers JM396 and JM403 were used for 5' copy-back DI-genome amplification and JM402 and JM396 – for MV full-length genome amplification (*Shingai et al., 2007*). (**B**) Comparison of immunostimulatory activities of RIG-I associated RNAs purified from MV and MVΔV infected cells. 10, 5, 2 and 1 ng of RNA were tranfected into STING-37cells, 5'3P, poly (I:C) and IFNβ (200 UI/mL) were used as controls and Fluc activity was measured after 24 hr post-transfection. Experiments were performed two times and data represents means ± SD of the technical triplicates of the most representative experiment. (**C**) RIG-I recognizes 5' copy-back DI-genome in MVΔV infected cells. ST-RIG-I cells were infected with MVΔV. RIG-I/RNA samples were subjected to Illumina strand-specific NGS. Reads were mapped to the MV genome and only the first nucleotide was retained in the X axis. Normalization and presentation of NGS results as *Figure 6*.

The following figure supplements are available for figure 7:

**Figure supplement 1.** The 5' copy-back DI genome of MVΔV.

**Figure supplement 2.** Comparison of immunostimulatory activities of total RNA purified from ST-RIG-I cells infected by either MV or MVΔV recombinant viruses.

## MV 5' copy-back DI-RNA specifically associates with RIG-I receptor and is efficiently purified using ST-RIG-I cells

The failure to detect any specific MV RNA ligands for RIG-I could be explained by the lack of RIG-I-specific MV RNA ligands, or by a deficiency of tagged RIG-I protein for RNA ligands purification. MV 5' copy-back DI genomes are known interactors of RIG-I. Therefore, we looked for the presence of DI RNA molecules in ST-RIG-I cells infected with MV by using universal 5' copy-back DI genome specific primers. RT-PCR analysis failed to detect any 5' copy-back DI RNA in infected ST-RIG-I cells (*Figure 7A*). Interestingly, when the same cells were infected with MV△V virus deficient in V protein, two DNA fragments of approximately 1 and 4 kb were amplified from total RNA samples using 5' copy-back DI genome specific primers (*Figure 7A*). Furthermore, sequencing analysis revealed that the 1 kb RT-PCR amplicon contained all the 5' copy-back DI-RNA features. Indeed, this 1236 nucleotide-long DI genome included genomic 'trailer' (Tr) and the reverse complement of the Tr sequences at the 3' and 5' extremities and the exact complementary of the extremities of 201 nucleotides able to hybridize to form a stem-loop structure (*Figure 7—figure supplement 1*). Sequencing analysis of the 4-kb amplicon suggested that it could correspond to a mosaic type of MV DI genome (*Marriott and Dimmock, 2010*). Further, the immunostimulatory activity of RIG-I/RNA purified from MV△V infected cells demonstrated a significant increase in ISRE stimulation compared to RIG-I/RNA purified from MV infected cells (*Figure 7B*). Interestingly, the immunostimulatory activity of total RNA purified from MV and MV△V infected ST-RIG-I cells was similar (*Figure 7—figure supplement 2*).

Since 5' copy-back DI genomes are known interactors of RIG-I, we used MVΔV infection to verify the efficiency of specific RIG-I ligands purification from ST-RIG-I cells. As above, RIG-I/RNAs were analysed by NGS. Again as a control, we used CH/RNA samples purified in a parallel experiment. As previously, the experiments were performed in triplicate, sequencing reads were mapped to MV genome and statistical analysis was performed. As expected, when normalized read counts were represented along MV genome, a vast majority of reads aligned to the 5'-end of the genome with sequences of both positive and negative polarities (*Figure 7C*). This 5' MV genome region perfectly corresponds to the 1236 nts-long 5' copy-back DI genome characterized above (*Figure 7—figure supplement 1*). These data clearly confirm the role of RIG-I in sensing 5' copy-back DI RNAs and suggest a role of the MV-V protein in controlling the DI genome's formation. They also demonstrate that ST-RIG-I cell line is a potent tool to purify specific RNA ligands of RIG-I upon infection.

## Primary and secondary structures analysis of RLR-specific viral RNA ligands

To evaluate AU/GC composition of the RNA sequences found to be specifically interacting with the three RLRs upon infection with MV, we performed additional bioinformatic analysis of the NGS data. AU content was calculated in a sliding window of 200 nts with one nucleotide step size and was compared to the mean count within that 200 nt window. *Figure 8A,B,C* demonstrates the AU composition of RIG-I/MDA5/LGP2 specific RNA partners. Our results show that RIG-I was preferentially binding AU-rich RNA regions of the MV genome, but only when the DI genomes were generated upon infection with MVΔV. Further, using Mfold algorithm we performed in silico analysis of the potential to form RNA secondary structures by the different MV genome regions. We observed that the 3'-end of the genome up to beginning of the M fragment possessed regions with lower Free Energy (*Figure 8D*).

## NGS analysis reveals that RIG-I recognizes specifically the AU-rich 3' region of CHIKV genome

Similar protocol of NGS analysis of RLR-specific RNA partners was applied in CHIKV-infected ST-RLR cells. Approximately, 16 million reads of 51 nucleotides in length were obtained from each sequencing sample and were mapped to the CHIKV genome. This gave viral genome coverage ranging from 597X to 15363X. We performed the same statistical analysis as for MV samples. Interestingly, this time total RNAs profiles of ST-RLR cells infected with CHIKV were different with respect to the negative control ST-CH cell line (*Supplementary file 1*). Again, these differences were taken into consideration upon statistical analysis. The distributions of normalized read counts matching the CHIKV genome sequences were represented along the viral genome with the X axis corresponding to all

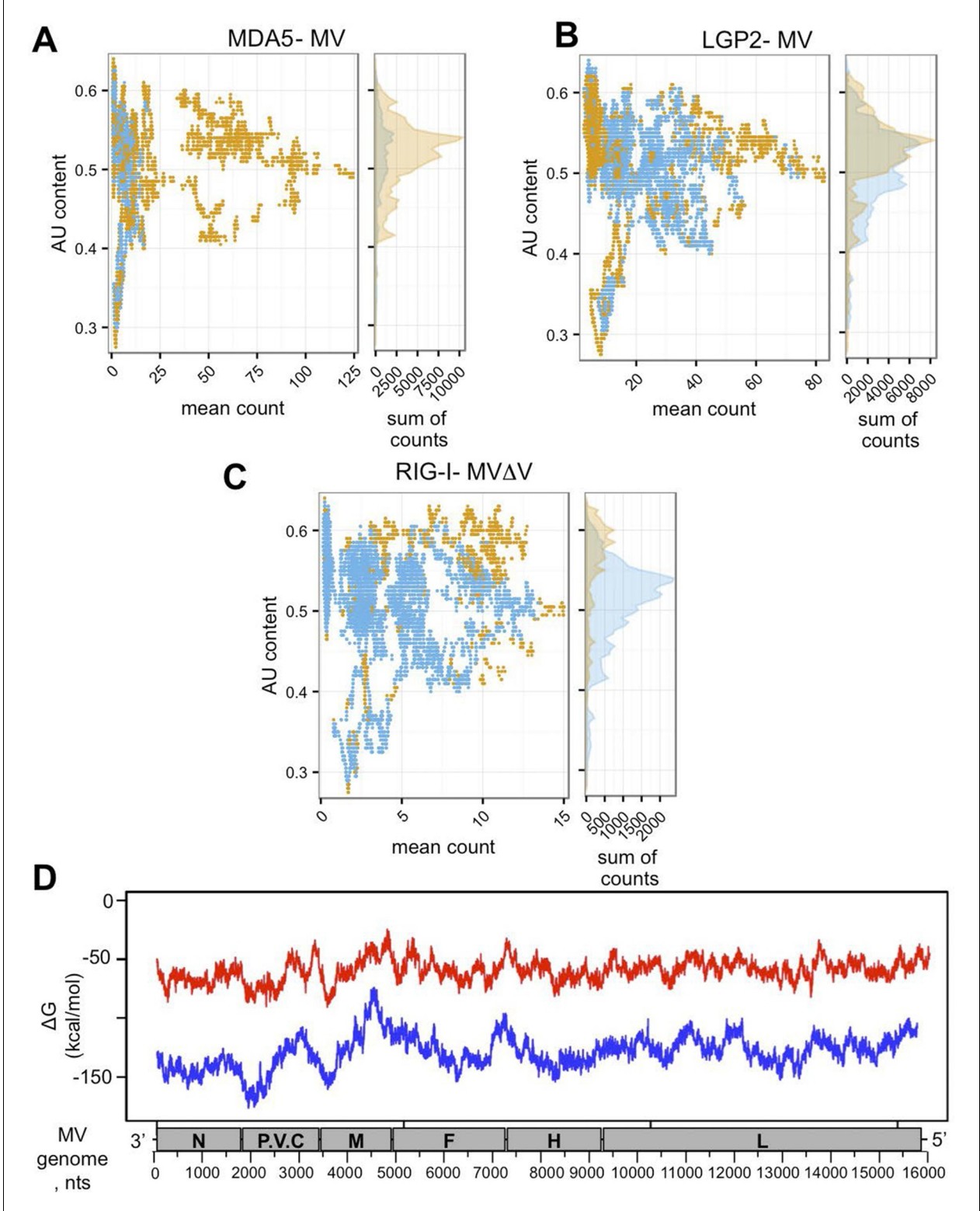

**Figure 8.** In silico analysis of NGS data. (A, B, C) AU content of RLR-specific RNA ligands. Number of sequenced reads (extended to 200nts) with a given AU content. Significantly enriched reads/positions are represented in orange and non-significantly enriched reads are coloured in blue. (A) MDA5, (B) LGP2, (C) RIG-I NGS data. (D) Secondary structure analysis of the MV genome. Either 250 (red) or 500 (blue) nucleotide long MV genome fragments were analysed. ΔG (free energy) *vs.* position on the MV genome is shown.

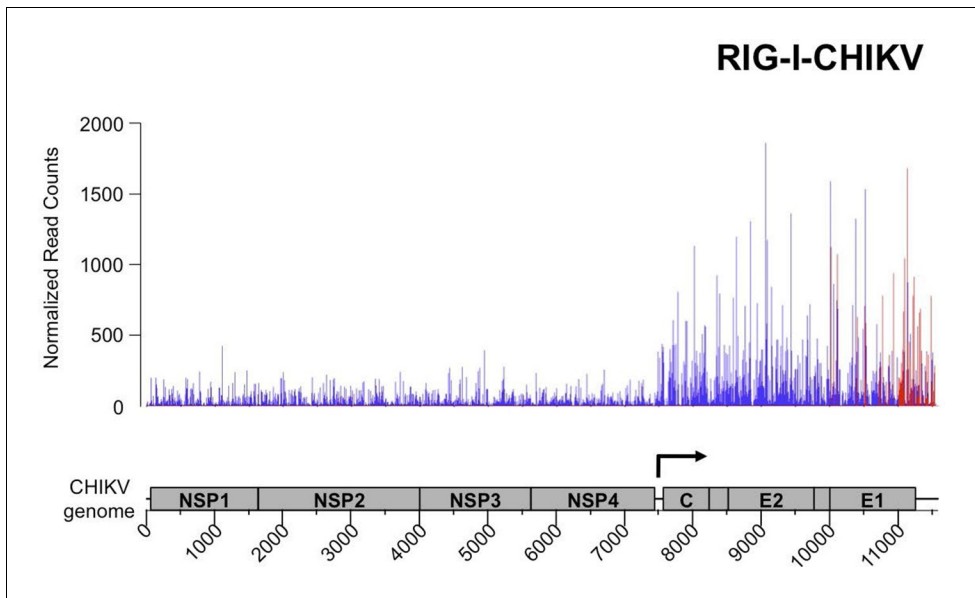

**Figure 9.** Analysis of purified RIG-I-specific RNA partners by NGS upon CHIKV infection. RIG-I/RNA samples were subjected to Illumina strand-specific NGS. Sequencing reads were mapped to the CHIKV genome and only the first nucleotide was retained in the X axis. Normalization and presentation of NGS results as *Figure 6*. The start of subgenomic RNA transcription is shown with the black arrow.

The following figure supplement is available for figure 9:

**Figure supplement 1.** NGS profile of total RNA aligned on the CHIKV genome from ST-CH cells infected with the CHIKV.

possible positions on the viral genome, and the Y axis showing the normalized number of reads that begin at that position (*Figure 9*).

First, we studied NGS data obtained for the total RNA samples. Visual inspection of the aligned reads on the CHIKV genome shows an enrichment of the C-E2-E1-coding region corresponding to the CHIKV subgenomic RNA (*Figure 9—figure supplement 1*). Indeed, previous studies have shown that alphaviruses subgenomic RNAs are more abundant than genomes in cytoplasm of infected cells (*Strauss and Strauss, 1994*; *Pushko et al., 1997*).

Although the average number of reads obtained for the CHIKV genome in all RLR samples was higher than for MV RLR/RNAs, after the statistical analysis LGP2/RNA and MDA5/RNA samples failed to show any specific enrichments in reads corresponding to the CHIKV genome (data not shown). In contrast, for the RIG-I/RNA sample we observed a specific enrichment in (+) sense RNA reads corresponding to the 3' untranslated region (3'-UTR) of the CHIKV genome (*Figure 9*).

Thus, our protocol of RLR-associated RNA ligands purification allowed us to observe that each of these cytosolic sensors has a specific RNA profile upon infection with different RNA viruses.

## Validation of NGS data using conventional approaches

We confirmed the relative amount of RLR-associated viral RNA by using qPCR. For this, numerous primers aligning on MV and CHIKV genomes were designed (*Figure 10A* and *Supplementary file 2*). The obtained enrichment in mRNA coding for MV-N protein in MDA5/RNA sample was compared to the CH/RNA sample. This qPCR analysis used three different probes aligning at the beginning and the end of the MV-N mRNA (*Figure 10A*) and validated that MDA5-associated RNAs most likely represent transcripts coding for the MV-N protein (*Figure 10B*). As negative control three other MV mRNAs coding for M, P and H proteins were looked for and not found to be enriched in MDA5/RNA samples (*Figure 10B*). We then validated that LGP2 specifically binds the 5'-end of the coding region of the N gene, as only the RNA fragment located the farthest to the 5'-end of the N mRNA (N1) was significantly enriched using qPCR (*Figure 10C*). Further, we validated the specific

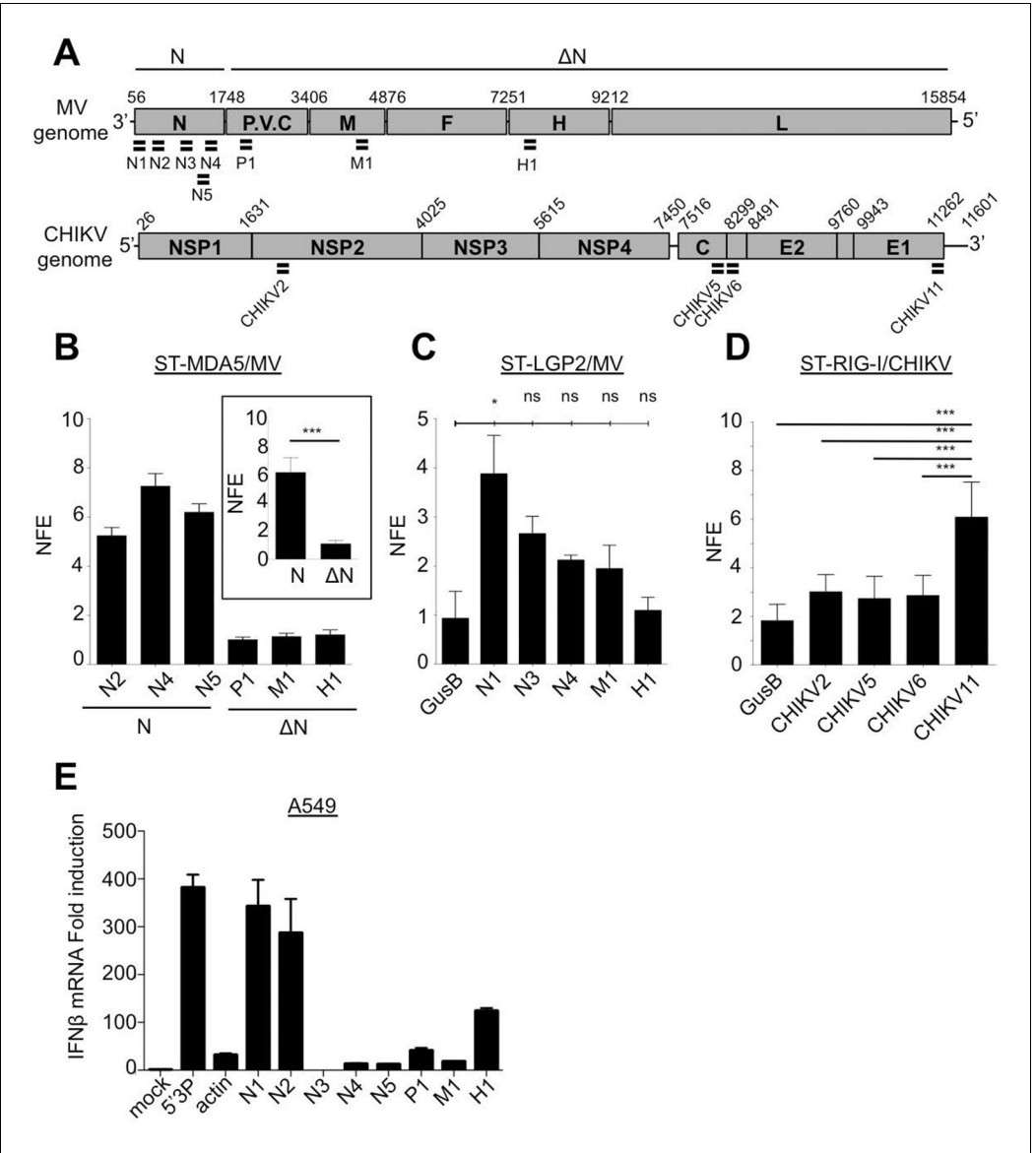

**Figure 10.** qPCR analysis of specific RLR RNA signatures from MV- and CHIKV-infected cells and their immunostimulatory activity. (**A**) Locations of qPCR primers on MV and CHIKV genomes. (**B**) MDA5 specific interaction with the N coding region upon MV infection. MDA5/RNA samples were subjected to RT-qPCR analysis with specific primers within N, P, M and H mRNAs. Relative Normalized Fold Enrichments (NFE) against the control CH samples are shown. Small window represents comparison of NFE of the amplicons obtained with set of primers located in the N region (N2, N4, N5) to the NFE of primers located elsewhere in the genome (P1, M1, H1). Comparisons were performed by a non-parametric Mann Whitney Test. (**C**) LGP2 specific interaction with the 5'-end of the N coding region upon MV infection.LGP2/RNA samples were subjected to RT-qPCR analysis with specific primers within N, P and M mRNAs and housekeeping gene GusB. Relative NFE against the control CH samples are shown. Fold enrichment for different primers were compared by One-Way-ANOVA and a Tukey Multiple Comparison test. (**D**) RIG-I specific interaction with the 3'-end of the CHIKV genome. RIG-I/RNA samples were subjected to RT-qPCR analysis with specific primers along CHIKV genome. Fold enrichment for different primers were compared by One-Way-ANOVA and a Tukey Multiple Comparison test (*p<0.05 ***p<0.001) (**E**) Immunostimulatory activity of in vitro transcribed RNA fragments corresponding to the RLR-specific regions on MV genome. RNA fragments were synthesized in vitro and transfected in A549 cells. IFNβ mRNA induction was measured by RT-qPCR analysis.

The following figure supplement is available for figure 10:

**Figure supplement 1.** co-IP of MV-N mRNA on MDA5 from human monocytes.

interaction of RIG-I with the 3'-end of CHIKV genome. For this, we used a pair of primers detecting only the 49S genomic RNA (CHIKV2), and three others detecting both the 49S genomic and 26S subgenomic RNAs (CHIKV5, 6, 11) of which only CHIKV11 was located at the 3'-end of the genome (*Figure 10A*). Indeed, only the primer located at the 3'-end of the genome was significantly enriched compared to the other sets of viral specific primers and to a cellular housekeeping gene (*Figure 10D*).

Furthermore, we analysed immunostimulatory activity of RLR-specific regions on MV genome by transfecting in vitro transcribed RNA fragments in A549 epithelial cells permissive to MV. We identified that only sequences at the 5'-end of the N mRNA had an enhanced capacity to activate the IFN response (*Figure 10E*).

Finally, we validated by classical co-immunoprecipitation (co-IP) approach the results obtained with our ST-RLR purification technique. For this we infected a typical virus-permissive immune cell model (acute monocitic leukemia THP-1 cell line) to perform MDA5-specific RNAs isolation by co-IP and RNA analysis by qPCR. This experiment validated that MDA5 has a predisposition to bind MV-N coding region (*Figure 10—figure supplement 1*).

These results are in good agreement with NGS results, confirming the relevance of our high-throughput protocol to study viral RNA signatures sensed by RIG-I, MDA5 and LGP2 cytosolic receptors upon infection with different RNA viruses.

## Discussion

RLRs are located at the frontline of the evolutionary race between viruses and the host immune system. These cytoplasmic proteins detect invasion of viral RNA in the cytoplasm of infected cells and trigger innate immunity. The inflammatory response is necessary for limiting viral replication and initiating a specific adaptive immune response. Most of the studies on RLR specific viral RNA ligands are performed in the absence of a productive virus infection (for review see [*Yoneyama et al., 2015*]). These studies are based on transfection within cells of RNAs coming from infected cells, expression of viral-coding sequences from plasmids, or transfection of in vitro transcribed RNA or synthetic RLR agonists such as poly(I:C). Recently, to characterize RNA ligands of RLRs in the presence of active viral infections (Sendai and Influenza viruses, then MV and EMCV), NGS has been applied on RNAs isolated from infected cells by co-IP. In these studies the methodological approach was based on either Photo Activable-Ribonucleoside-enhanced CrossLinking and co-IP (PAR-CLIP) or simple co-IP followed by NGS. Using these approaches for Sendai and Influenza viruses only RIG-I specific RNA partners were analysed (*Baum et al., 2010*), for EMCV only LGP2-specific RNA partners were sequenced (*Deddouche et al., 2014*), and for MV RIG-I and MDA5 specific RNA partners were studied simultaneously (*Runge et al., 2014*). The disadvantages of RNA-protein isolation by co-IP approaches are that (i) they rely on the availability of high-affinity antibodies against the protein of interest and (ii) they often lack an appropriate negative control. Negative control antibodies with different affinities and directed against proteins that are absent from the cells can easily underestimate the background noise generated by co-IP followed by NGS. Also, for the PAR-CLIP methodology, the incorporation of the nucleoside analogue 4-thiouridine (4SU) within RNA during the active cycle of viral replication could change the molecular properties of viral and cellular RNAs, thus creating an artificial environment.

To overcome these problems we applied One-STrEP-tag fusion protein affinity chromatography to purify RLRs. For this, we generated three single-clone stable cell lines expressing either tagged RIG-I, or MDA5, or LGP2 (*Figure 1*). In addition, a stable cell line expressing CH protein fused to the same One-STrEP-tag was used as a control for non-specific binding. These tools allow studying the three RLRs in the same setting and to characterize their specific viral RNA ligands in the course of infection. Indeed, we isolated specific RLR RNA ligands from these cells infected with MV and CHIKV, respectively a negative- and a positive-sense RNA virus. Three biological replicates were performed for each RLR and an appropriate negative control (CH/RNA) allowed subtracting non-specific binding and to perform rigorous statistical analyses.

Despite the fact that these cells over-expressed one RLR, they were susceptible to infection with different RNA viruses (*Figures 2* and *3*). By applying multiple functional tests we determined the best conditions for ST-RLR cells infection (*Figure 3C*). We validated that they were efficient for RLR-specific RNA ligands purification and allowed extraction of immunoactive RNA molecules (*Figure 4*).

Consistent with previous studies we observed that RIG-I/RNAs provided a stronger immunostimulatory activity in comparison to MDA5/RNA molecules (*Ikegame et al., 2010*; *Runge et al., 2014*). In the same experimental set-up we showed that LGP2-specific RNA had a similar immunostimulatory activity to RIG-I/RNAs (*Figure 4*). Thus, for the two infections more immunoactive RNA ligands were purified with LGP2 and RIG-I than with MDA5.

For CHIKV infection, specific RNA ligands of cytosolic RLRs have not been established yet. The inflammatory response in non-hematopoietic cells is known to be dependent on MAVS, the adaptor molecule necessary for both RIG-I and MDA5, however which of the two receptors is involved in sensing CHIKV is still unknown (*Schilte et al., 2010*). We found that RIG-I specifically interacts with the 3'-UTR of CHIKV genome. Despite the higher coverage of the CHIKV genome compared to MV genome in our NGS data, we failed to observe specific enrichment in CHIKV reads on LGP2 and MDA5 receptors. The observation that RIG-I specifically interacts with CHIKV 3'-UTR is very promising (*Figure 9*). Indeed, the 3'-UTR of positive-sense RNA viruses has already been shown to be particularly immunoactive (*Saito et al., 2008*; *Schnell et al., 2012*; *Kell et al., 2015*). CHIKV 3'-UTR contains multiple 21 nucleotide-long repeated sequence elements forming dsRNA structures, which are possible targets for RIG-I (*Zhang et al., 2013b*). Interestingly, the 3'-UTR has recently been described as a region particularly sensitive to the evolutionary pressure exerted by mosquito vectors (*Chen et al., 2013*). Thus, we observed that in the mammalian host RIG-I targets the same viral region. To our knowledge this is the first time that RIG-I affinity for the 3'-end of a positive-sense RNA virus is demonstrated in physiological conditions.

For MV infection we compared our NGS data with those previously published for RIG-I and MDA5 using the PAR-CLIP coupled to NGS approach (*Runge et al., 2014*). Interestingly our study and the Runge *et al.* data showed that i) RLRs preferentially interact with MV-N and MV-L derived RNA, ii) RIG-I and not MDA5 interacts with viral RNA of both negative- and positive-sense polarities originating from MV 5' copy-back DI genomes (*Figure 7C*, [*Runge et al., 2014*]). However, we observed several discrepancies. We found that MDA5 and not RIG-I specifically interacted with MV-N coding sequences. RIG-I specific positive- and negative-sense reads only represented 5' copy-back DI-genome and were equally distributed along the 5'-end of the MV genome (*Figure 7C*). Similar NGS profile of the 5' copy-back DI genome on RIG-I was previously observed in the context of Sendai virus infection (*Baum et al., 2010*). In this study only a single population of 546 nts-long 5' copy-back DI RNA genome has been specifically enriched on RIG-I. Thus, the concordance of our results for RIG-I with Baum et al. study nicely confirms the relevance of our strategy.

In the future, a deeper NGS study should be done to explain why the RIG-I/RNA sample collected during MV infection possesses immunostimulatory activity despite the absence of MV RNA ligands specifically bound to RIG-I (*Figure 4A* and *6C*). We propose that self-RNA molecules processed by the action of the antiviral endoribonuclease RNaseL on cellular RNA could be specific RNA ligands that bind RIG-I and induce IFN-β expression. Indeed, RNaseL induces type I IFN *via* production of short-length RNA molecules of viral and cellular origins by the mechanism that includes interaction with MDA5 and RIG-I (*Malathi et al., 2007*; *2010*). In the current experimental setup we used a sample multiplexing that allowed an appropriate analysis of relatively long viral-derived RNA and not cellular RNA molecules. Alternatively, it is possible that MV RIG-I-specific RNA ligands are short length and thus not compatible with the classical Illumina NGS protocol applied in this study. In the future this protocol will be adapted to study small size RLR partners and to study RLR-specific cellular partners.

Recently Deddouche *et al.* have isolated a pertinent MDA5 agonist by co-IP of LGP2/RNA complexes purified from cells infected with EMCV, a positive-sense RNA virus. Using NGS, they mapped this RNA on the L region of the EMCV genome. Numerous functional assays were performed to validate that the L region is a key determinant of the MDA5 stimulatory activity (*Deddouche et al., 2014*). This work supported previous studies showing that in an infected cell, MDA5 and LGP2 positively cooperate with each other (*Venkataraman et al., 2007*; *Satoh et al., 2010*; *Bruns and Horvath, 2012*; *Childs et al., 2013*). In vitro studies of MDA5 suggested that MDA5 hypothetically recognizes long dsRNA molecules like poly(I:C) (*Kato et al., 2008*). MDA5 binds dsRNA from the stem of the molecule and starts forming filaments towards its extremities (*Peisley et al., 2011*; *Wu et al., 2013*). Recent studies of autoimmune disorders have proposed that MDA5 detects aberrant secondary structures formed by self mRNAs (*Liddicoat et al., 2015*). All these are evidences that most certainly MDA5 natural ligands are RNA stem structures embedded within different RNAs.

In concordance with this, our results show that MDA5 recognizes the integrity of the MV-N mRNA, certainly through the recognition of stem loop structures. Further, our in silico (Mfold) prediction of RNA secondary structures within the different parts of the MV genome suggested that MV-N mRNA had a potential to form stem loop structures (*Figure 8D*).

It has recently been shown that LGP2 can stabilize shorter filaments of MDA5 around dsRNA of 100 nts in length (*Bruns et al., 2014*). Our data further confirmed this MDA5/LGP2 synergy. We demonstrated that upon MV infection MDA5 and LGP2 were binding similar viral RNA ligands. Indeed, these LGP2 specific RNAs were encompassing the MV-N segment (*Figure 6A and B*). However, while the whole MV-N mRNA was bound by MDA5, the LGP2-specific reads were mostly localized in the 5'-end of the N region. Furthermore, when transfecting in vitro transcribed RNAs corresponding to the MDA5- and LGP2-specific regions of the MV genome in human epithelial cells, we identified a higher immunostimulatory activity of the 5'-end of the N mRNA compared to RNA fragments of similar size but coming elsewhere from the genome (*Figure 10E*). These data suggest that, upon MV infection, LGP2 and MDA5 specifically interact with the same RNA agonist and that LGP2 could be involved in stabilizing short MDA5 filaments formed on secondary structures localized at the 5'-end of the MV-N mRNA.

In summary, using One-STrEP-RLRs affinity chromatography purification and NGS we provide the first simultaneous visualisation of specific RNA ligands for RIG-I, MDA5 and LGP2 in living cells and in the presence of different RNA viruses. Our results show that each of these cytosolic sensors has its individual RNA profile upon infection with different RNA viruses (*Figures 6*, *7* and *9*). Previous studies demonstrated RIG-I specific binding to the AU- and polyU/UC-rich regions of viral genomes (*Saito et al., 2008*; *Schnell et al., 2012*; *Runge et al., 2014*; *Kell et al., 2015*). In our results RLR-specific binding cannot simply be explained by primary RNA composition or AU/GC content of RNA ligands. Indeed, we found RIG-I preferential binding to the AU-rich sequences (*Figure 8C*), but this binding was observed only when a dsRNA-containing DI genome was produced by the recombinant MV△V. These results suggest that RIG-I recognizes specific RNA secondary structures. MDA5 and LGP2 specificity to AU-rich RNAs was less pronounced (*Figure 8A and B*). Additional studies should be performed to obtain a clearer view on the similarities and the differences of RNA structures recognized by each of the three RLRs. For this, we are currently applying our One-STrEP-RLR approach to three other RNA viruses as well as to an intracellular bacterium. As a consequence, this work offers new strategies for immune and antiviral therapies by targeting the RLR pathway for the therapeutic control of viral infections, enhancing the immune response for vaccines, and conceiving strategies for immune suppression to control inflammation or specific autoimmune diseases.

# Materials and methods

## Cells, plasmids and recombinant viruses

HEK-293 (human embryonic kidney, ATCC CRL-1573), Hela (human cervix adenocarcinoma epithelial, ATCC CCL-2) and A549 cells (human lung adenocarcinoma epithelial, ATCC CCL-185) were maintained in DMEM-Glutamax (GIBCO, Thermo Fisher Scientific, Waltham, Massachusetts) supplemented with 10% heat-inactivated FCS (Invitrogen, Thermo Fisher Scientific, Waltham, Massachusetts) and 100 U/ml/100 µg/ml of Penicillin-Streptomycin (GIBCO). THP-1 (monocyte cells, ATCC TIB-202) were cultured at $0.5–7 \times 10^5$ cells/ml in RPMI 1640 (GIBCO) containing 10% heat-inactivated FCS, 100 U/ml/100 µg/ml of Penicillin-Streptomycin. For all cell lines mycoplasma contamination testing status was routinely verified (#30-1012K, ATCC). To establish clonal stable cell lines expressing One-ST-RLR, we used a modified pEXPR-IBA105 plasmid carrying GW cassette (pEXPR-IBA105GW) kindly provided by Dr. Yves Jacob (Unité de Génétique Moléculaire des Virus ARN, Institut Pasteur). First, the RLR sequences (LGP2, MDA5 or RIG-I) were amplified by standard PCR from a human spleen cDNA library (Invitrogen) using specific primers with AttB1 and AttB2 sequences included (*Supplementary file 2*). The corresponding DNA fragments were cloned by in vitro recombination into pDONR207 entry vector (BP reaction, Gateway, Invitrogen). RLR genes (MDA5: AF095844; LGP2: AAH14949 RIG-I: CCDS6526.1) were finally recombined from pDONR207 to One-STrEP-tag pEXPR-IBA105GW by in vitro recombination (LR reaction). These new plasmids were transfected into HEK-293 cells using JetPrime reagent (#114–15, Polyplus Transfection, Strasbourg, France). Two days later, culture medium was removed and replaced by fresh medium containing

G418 at 500 µg/ml (#G8168, SIGMA, St. Louis, Missouri). Transfected cells were amplified and subsequently cloned by serial limit dilution. At least 5 clones were screened for each RLR to detect the tagged protein expression by qPCR and Western Blot.

For STING37shRLR cell lines, we generated lentiviral vectors using the canonical triple transfection of HEK293T cells by a VSVg envelope, an encapsidation p8,74 plasmid kindly provided by Dr. Pierre Charneau (*Zufferey et al., 1997*) and a pGIPZ vector plasmids (Thermo Scientific, Waltham, Massachusetts) expressing either an shRNA with no target (RHS4346, shNeg), or targeting LGP2 (shLGP2, RHS4430-99166661-V2LHS_116526), MDA5(RHS4430-101128286-V3LHS_300657, shMDA5) or RIG-I (RHS4430-99619609-V2LHS_197176, shRIG-I), all bearing a puromycin selection marker. Vectors were titrated according to manufacturer's instructions in HeLa cells. ISRE reporter cell line (STING37 [*Lucas-Hourani et al., 2013*]) was transduced at an MOI of 0.3 and 48 hr later puromycin (5 µg/ml) was added to the media to select properly transduced cells.

CHIKV 06–49 strain was isolated from the serum of an adult patient with arthralgia/ myalgia in Saint Louis city, Réunion, France in December 2005 (*Schuffenecker et al., 2006*). CHIKV strain 06–49 was titrated on VERO cell by TCID50 (50% Tissue Culture Infective Dose). Recombinant CHIKV-Rluc which contains the *Rluc* reporter gene inserted between the non-structural nsP3 and nsP4 proteins has already been described (*Henrik Gad et al., 2012*). Attenuated MV Schwarz vaccine strain (MV) and recombinant MV Schwarz expressing *Fluc* (rMV2-Fluc) from an additional transcription unit derived from MV have been previously described (*Combredet et al., 2003*; *Komarova et al., 2011*). To prevent V protein expression from MV, a two-step PCR strategy was used to generate MVΔV virus. Two PCR fragments were amplified using MV2313 (5'-ATCTGCTCCCATCTCTATGG) and MV2504 (5'-TCTGTGCCCTTCTTAATGGG) for the first fragment and MV2485 (5'-CCCATTAA-GAAGGGCACAGA) and MV3385 (5'-AGGTTGTACTAGTTGGGTCG) for the second fragment. These PCRs introduced a mutation interfering with RNA editing (UUAAAAAGGGCACAGA native sequence was mutated to UUAA**g**AAGGGCACAGA). The two PCR products were combined in a second PCR reaction using MV2313 and MV3385 primers. The produced mutated HindIII-SpeI MV fragment was moved into the pTM-MVSchwarz plasmid after digestion with the corresponding restriction enzymes. Recombinant virus was rescued as previously described (*Combredet et al., 2003*) and named MV△V. Immunoblot analysis was performed to validate the lack of MV-V expression by MVΔV (data not shown). Virus stocks were produced on VERO cells, and titrated by TCID50.

## Antibodies, western blot and intracellular staining

Protein extracts were resolved by SDS-polyacrylamide gel electrophoresis on 4–12% Criterion gels (BioRad, Hercules, California) with MOPS running buffer and transferred to cellulose membranes (GE Healthcare, Little Chalfont, United Kingdom) with the Criterion Blotter system (BioRad). The following antibodies were used: an anti-STrEP-Tag (#34850, Qiagen, Hilden, Germany), anti-LGP2 (NBP1-85348, Novus, Littleton, Colorado), anti-MDA5 (#5321, Cell Signaling, Danvers, Massachusetts and AT113, EnzoLifescience, New York, NY), anti-RIG-I (D14G6, Cell Signaling) or monoclonal anti-β-actin antibody (A5441, Sigma), IRF3 (#11904, Cell Signaling) or phosphor-IRF3 (ab76493, Abcam, Cambridge, UK,). HRP-coupled anti-mouse (NA9310V, GE Healthcare) or anti-rabbit (RPN4301, GE Healthcare) were used as secondary antibodies. Peroxidase activity was visualized with an ECL Plus Western Blotting Detection System (#RPN2132, GE Healthcare). MV intracellular staining was performed with mouse anti-N mAb (clone 25, [*Giraudon and Wild, 1981*]) and FITC coupled Goat Anti-mouse Ab (BD Biosciences, Franklin Lakes, New Jersey). For CHIKV, intracellular staining was performed with either FITC-conjugated anti-CHIK.E2 mAB 3E4 (*Bréhin et al., 2008*) or anti-dsRNA mAb (J2-1201, Scicons, Szirák, Hungary) followed by anti-mouse-APC antibody (A865, Invitrogen).

## MV and CHIKV replication assays

STING37shRLR cells (2 x 10$^5$ per well) were plated in a 24-well plate and infected with appropriate MOIs. For rMV2-Fluc or CHIKV-Rluc infection, at each time point cells were lysed by Passive Lysis buffer (E1941, Promega, Fitchburg, Wisconsin), and luciferase activity was measured with Bright-Glow Luciferase assay system (E2650, Promega) or Renilla Glow Luciferase Assay System (E2720, Promega), correspondingly. For immunostaining, cells were washed twice with phosphate-buffered saline (PBS) and 2% foetal calf serum (FCS) and then fixed in PBS containing 4% paraformaldehyde. Cells were permeabilized with PermWash buffer (#554723, BD Biosciences), incubated with the

primary antibody at 4°C for 30 min, washed in PermWash and incubated with the secondary antibody. Cells were washed twice with PBS 2%FCS before analysis by flow cytometry using a MACS-Quant cytometer (Miltenyi Biotec, Bergisch Gladbach, Germany) and analysis was done with the software FlowJo (vers 7.6).

## Affinity chromatography of RLR RNP complexes and subsequent RNA and protein purification

ST-RLR cells ($5\times10^7$) were infected at an MOI of 1 for 24 hr (MV) or for 13 hr (CHIKV). Cells were washed twice with cold PBS and lysed in 6 ml of lysis buffer (20 mM MOPS-KOH pH7.4, 120 mM of KCl, 0.5% Igepal, 2 mM $\beta$-Mercaptoethanol), supplemented with 200 unit/ml RNasin (#N2515, Promega) and Complete Protease Inhibitor Cocktail (#11873580001, Roche, Penzberg, Germany). Cell lysates were incubated on ice for 20 min with gentle mixing every 5 min, and then clarified by centrifugation at 16,000 g for 15 min at 4°C. A 250 µl aliquot of each cell lysate was used to perform total RNA purification using TRI Reagent LS (#T3934, SIGMA). The remaining of cell lysate was incubated for 2 hr on a spinning wheel at 4°C with 200 µl of StrepTactin Sepharose High Performance beads (# 28-9355-99, GE Healthcare). Beads were collected by centrifugation (1600 g for 5 min at 4°C) and washed twice for 5 min on a spinning wheel with 5 ml of washing buffer (20 mM MOPS-KOH pH7.4, 120 mM of KCl, 2 mM $\beta$-Mercaptoethanol) supplemented with 200 unit/ml RNasin and Complete Protease Inhibitor Cocktail. Precipitates were eluted using biotin elution buffer (IBA). RNA purification was performed using TRI reagent LS (T3934, SIGMA). RNA was dissolved in 80 µl of DNase-free and RNase-free ultrapure water. Extracted RNAs were analyzed using Nanovue (GE Healthcare) and Bioanalyser RNA nano kit (#5067–1511, Agilent, Santa Clara, California).

## Next generation sequencing (NGS)

RNA molecules isolated from ST-RLR/RNA complexes were treated for library preparation using the Truseq Stranded mRNA sample preparation kit (Illumina, San Diego, California,) according to manufacturer's instruction. To analyze all RNA species present, the initial poly(A) RNA isolation step was omitted. Briefly, the fragmented RNA samples were randomly primed for reverse transcription followed by second-strand synthesis to create double-stranded cDNA fragments. No end repair step was necessary. An adenine was added to the 3'-end and specific Illumina adapters were ligated. Ligation products were submitted to PCR amplification. Sequencing was performed on the Illumina Hiseq2000 platform to generate single-end 51 bp reads bearing strand specificity.

## Read mapping, expression quantification and statistical analysis

Sequenced reads have been cleaned from adapter sequences and low complexity sequences (see appendix). Reads were aligned to the MV Schwarz strain reference genome (http://www.ncbi.nlm.nih.gov/nuccore/FJ211590.1), and to CHIKV49 strain reference genome (http://www.ncbi.nlm.nih.gov/nuccore/AM258994) using bowtie (version:0.12.7, options: -a –best -q -m50 -e50 –chunkmbs 400 [*Langmead et al., 2009*]). The first position of each read (taking into account the strandness) was used for statistical analysis using the KNIME software (*Jagla et al., 2011*). Counts per position where calculated using pileup from Samtools (*Li et al., 2009*).

Reads cleaning was performed using version 0.10 of clean_ngs (https://github.com/PF2-pasteur-fr/clean_ngs) in single-end mode and the following parameters: minLen=15, maxLen5 3' threshold = 5' threshold = 61 (ASCII value). Adapter sequences: TGGAATTCTCGGGTGCCAAGG; TGGAATTCTCGGGTGCCAAGGAACTCCAGTCACNNNNNNNATCTCGTATGCCGTCTTCTGCTTG; parameters for both sequences: Identity 0.85 (min identity between adapter and read), QC threshold= 61 (ASCII value, bases with values inferior are considered as a match to the adapter, remove as many adapter as possible) min overlap=7; truncate adapter = 0 (don't truncate adapter at 5'-end); leader sequence = 0 (use adapter sequence as a regular adapter at the 3'-end).

Statistical analyses were performed with R version 3.0.2 (*Team, 2013*) and bioconductor packages. Normalization and differential analyses were carried out between RLR/RNA and Ch/RNA on one hand, and between total RLR RNA and total CH RNA on the other hand. In both cases the normalization was performed with DESeq2 version 1.2.1 with the normalization parameter locfun="shorth" and default parameters otherwise (*Love et al., 2014*). For the differential analysis we used DESeq2 with default settings. Raw p-values were adjusted according to the BH procedure

(*Benjamini and Hochberg, 1995*). Adjusted p-values were considered significant if they were lower than 0.05. Positions of interest were those enriched in reads in RLR/RNA compared to CH/RNA, and not enriched in the total RNA comparison.

## In silico analysis of NGS data

AU content was calculated in a sliding window of 200 nts (step size=1) and compared to the mean count within the 200 nt window and to the sum of sequenced reads (extended to 200 nt). Viral RNA sequences were subjected to secondary structure prediction using Mfold software (version 3.6, MAX=1) using standard parameters to all sliding windows (window size= (250 and 500); step size=1) from the reference genome.

## RT-PCR detection and cDNA cloning of 5' copy-back DI-RNAs

The copy-back DI-RNAs were amplified from RNA extracted from total RNA extracted from ST-RIG-I using two sets of MV primers (*Shingai et al., 2007*): for DI-RNAs, 396 (A: 5'-TATAAGCTTACCAGA-CAAAGCTGGGAATAGAAACTTCG) and 403 (C: 5'-CGAAGATATTCTGGTGTAAGTCTAGTA) and for standard genome, 396 (A) and 402 (B: 5'-TTTATCCAGAATCTCAAGTCCGG). Reverse transcription was performed with Super Script III (18080–0440–044, Invitrogen) and a PCR amplification was achieved with Q5 High-Fidelity 2X Master Mix (M0494S, NEB, Ipswich, Massachusetts). The PCR-amplified product A-C was cloned into pTOPO vector (Invitrogen) and sequenced.

## TaqMan RT-qPCR and SYBR Green qPCR analyses

RLR mRNA expression profile in ST-RLR cells was performed by RT-qPCR on total RNA isolated with the RNeasy Mini Kit (Qiagen). Reactions were performed with 100 ng of RNA using TaqMan RNA-to-Ct 1-Step Kit (# 4392938, Applied Biosystems, Waltham, Massachusetts) and 1 μl of custom Taq-Man Gene Expression Assays (Hs00920075_m1 - for LGP2; Hs01070332_m1 – for MDA5; Hs00204833_m1 – for RIG-I; Hs99999905_m1 for GAPDH; Hs01077958_s1 – for IFN-β). We used Applied Biosystem StepOnePlusTM system. Results were normalized using expression levels of the GAPDH housekeeping genes and RLR expression level in HEK293 cells was settled as 100% of gene expression. All the measures were performed in triplicate and analysed by StepOnePlusTM software. For RLR/RNA enrichment profile, RNA was extracted with TRI Reagent LS before or after affinity chromatography purification on StrepTactin Sepharose. Starting from 200 ng of RNA, cDNA synthesis was achieved in 20 μL using the SuperScript VILO cDNA Synthesis Kit (#1648108, Life Technologies, Thermo Fisher Scientific, Waltham, Massachusetts) following manufacturer's recommendations. Reactions were performed on 2 ng of cDNA using Power SYBR® Green PCR Master Mix (#4367659, Life Technologies). Reactions were performed in a final volume of 20 μl in the presence of 0,6 μM of appropriate primers (*Supplementary file 2*). MV and CHIKV specific primers were designed using Primer Express Software (Applied Biosystems). RNA Fold Enrichment was calculated according to the following formulas:

Fold Enrichment (FE):

$$FE(Ct) = 2^{-[(Ct(query) - Ct(GAPDH))beads] - [(Ct(query) - Ct(GAPDH)) total]}$$

Normalized Fold Enrichment (NFE):

$$NFE(primer) = \frac{RLR.FE(primer)}{mCherry.FE(primer)}$$

## Analysis of ST-RLR/RNA immunostimulatory activity

Short 5'3P-bearing RNA molecules were obtained from pCIneo plasmid linearized by XbaI before transcription. T7 transcription reactions were carried out with a T7 RiboMAX Express Large Scale RNA Production System (# P1320, Promega). RNA was purified using TriLS reagent and analyzed with Bioanalyser RNA nano kit (Agilent). Poly(I:C) was from Amersham Biosciences (# 27-4729-01, Amersham, UK). To determine ST-RLR responsiveness to synthetic ligands, expression of IFN-β was determined by transient transfection of reporter plasmid pIFNβ-Fluc containing the *Fluc* gene under the control of the IFN-β promoter (IFN-b-pGL3 [*Lin et al., 2000*]). For RNA transfection analysis, ST-RLR cells were plated in 24-well plates (2 x $10^5$ per well). After 24 hr, cells were transfected with

JetPrime reagent with pIFNβ-Fluc (250 ng/well), plasmid harbouring a thymidine kinase (Tk) promoter just upstream of the *Rluc* gene (pTK-Rluc, 25 ng/well) and 10 ng of either poly(I:C) or 5'3P. After 24 hr, cells were lysed, and the Fluc and Rluc activities were measured in cell lysates using Dual-luciferase Reporter Assay System (Promega) according to manufacturer's instructions. Reporter activity was calculated as a triplicate of the ratio of Fluc activity to the reference Rluc activity.

To determine the immunostimulatory activity of RLR RNA partners, STING-37 cells were plated one day before transfection in 24-well ($2 \times 10^5$ cells per well). RLR-specific or total RNA (10 ng) was transfected using jetPRIME together with 250 ng of pcINEO empty vector as a carrier. 24 hr post-transfection, cells were lysed with passive buffer and the Fluc activity was measured using the Bright-Glo Luciferase Assay System.

Viral sequences were synthesized using specific primers (*Supplementary file 2*) with the forward primers presenting the T7 promoter sequence (attgtaatacgactcactataggg) at the 5'-extremity. Amplicons were purified in 1% agarose gel and used for in vitro transcription. Desired RNA molecules were further separated from the 3'extension by-products (*Triana-Alonso et al., 1995*) on 5% TBE-Urea Polyacrylamide Gel (#345–0086, Biorad). Purified RNAs were analyzed with Bioanalyser RNA pico kit (Agilent). 10 ng of purified RNA was transfected in A549 cells and IFN-β mRNA was quantified by RT-qPCR analysis.

## Immunoprecipitation of MDA5-associated RNA from virus-infected cells

Similar to Runge et al. immunoprecipitation protocol was used but with little modifications: without crosslinking and the 20 mM MOPS-KOH pH7.4, 120 mM of KCl, 0.5% Igepal, 2 mM β-Mercaptoethanol supplemented with 200 unit/ml RNasin and Complete Protease Inhibitor Cocktail buffer was used for cell lysis and washing steps.

## Acknowledgements

We thank Dr. Yves Jacob and Dr. Pierre Charneau for providing material and Charlotte Romanet for technical support. The authors thank all the members of the Viral Genomics and Vaccination research unit for their help and useful discussions.

## Additional information

### Funding

| Funder | Grant reference number | Author |
|---|---|---|
| Fondation pour la Recherche Médicale | FDT20140931129 | Raul Y Sanchez David |
| Agence Nationale de Recherches sur le Sida et les Hepatites Virales | CSS1-AO 2012-2 | Frédéric Tangy Anastassia V Komarova |
| Agence Nationale de Recherches sur le Sida et les Hepatites Virales | CSS1-AO 2012-1 | Anastassia V Komarova |

The funders had no role in study design, data collection and interpretation, or the decision to submit the work for publication.

### Author contributions

RYSD, OS, AVK, Conception and design, Acquisition of data, Analysis and interpretation of data, Drafting or revising the article; CC, Conception and design, Acquisition of data, Analysis and interpretation of data, Contributed unpublished essential data or reagents; M-AD, BJ, PD, FT, Conception and design, Analysis and interpretation of data, Drafting or revising the article; J-YC, Conception and design, Analysis and interpretation of data; MM, Acquisition of data, Analysis and interpretation of data; MGG, Acquisition of data, Contributed unpublished essential data or reagents

Author ORCIDs

Anastassia V Komarova, http://orcid.org/0000-0002-9998-8447

## Additional files

### Supplementary files

• Supplementary file 1. Number of positions within the MV or CHIKV genomes different in total RLR RNA samples with respect to the negative control CH sample.

• Supplementary file 2. List of primers used for the study.

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
