## [Decision Letter]

Thank you for choosing to send your work, "Comparative analysis of viral RNA signatures on different RIG-I-like receptors", for consideration at *eLife*. Your submission has been assessed by Tadatsugu Taniguchi (Senior editor), Xuetao Cao (Reviewing editor) and two reviewers.

Although the work is of interest and significance, both reviewers have raised questions regarding your work. We recommend that further analysis of your data may be required; especially how to distinguish your findings from the other similar researches and what are the new concepts revealed by your study. Reviewer 1 has raised concerns about the presentation of the data, and Reviewer 2 has raised concerns about the cell model, binding profile of individual RLRs (common features and differences) and confirmation of stimulatory RNAs. Taking together these comments, we suggest that you may need to finish the following revisions.

1) Please provide additional information of the present data as suggested by the two reviewers.

2) We understand that it is inconvenient to perform further analysis in a new cell model. So it's not necessarily required by the journal to perform other NGS assays.

3) Further bioinformatic assays of the current data are helpful to understand the commons and differences of RNAs recognized by individual RLRs. At least, further description of the detected RNA profiles and the comparison to previous studies is necessary.

4) For the confirmation of the detected RNA signatures, it can be performed in alternative permissive cells that naturally respond to the viruses using the enriched RNAs.

The detailed comments of the two reviewers are attached below.

*Reviewer 1:*

In this manuscript, the authors showed that the negative- and positive-sense RNA genome signatures that were caused by RIG-I-like receptors (RLRs) activation. It is known that three RLR members (RIG-I, MDA5, and LGP2) are playing the cytosolic receptor to response for RNA viruses. However, their individual biological ligands are not clear and the molecular mechanism between RLRs and the associated viral RNAs is also unknown. To address this issue, the authors established the One-STrEP-tagged RLRs assay system. This system is well designed and has not disturbed the biological functions of RLRs in stable cell lines. The authors also optimized the best time points to purify the RLR-associated viral RNA ligands under various conditions. The NGS study showed that the MDA5 recognizes the positive strand of MV N region and LGP2 recognizes both stands of L region. For CHIKV infection, RIG-I accesses both strands of 3'UTR in viral genome. The whole data set reveals how RLRs recognize the viral RNA ligands during infection.

My major concern is the NGS analysis data. The significant enriched reads represented in red and non-significant reads in blue from Figure 6 to Figure 8. But the normalized read counts are not many in L region of LGP2-MV in Figure 6. The scale of the y-axis is shorter than MDA5-MV, thus how is the specificity of each region represented in this data? In Figure 7, where is the position of V region of MVΔV? In Figure 8, there are many enrichment reads around C/E2 region in RIG-I-CHIKV sample. What is the biological meaning of these differences?

*Reviewer 2:*

In the article entitled "Comparative analysis of viral RNA signatures on different RIG-I-like receptors", Dr. Komarova and colleagues developed a high-throughput riboproteomic approach based on One-STrEP-tagged protein affinity purification and NGS. Using this approach they compared RNA signatures in MV and CHIKV that could specifically be recognized by RIG-I, MDA5 and LGP2. They observed that 1) RIG-I recognized defective interfering genomes whereas MDA5 and LGP2 specifically bound MV nucleoprotein coding region and 2) RIG-I associated specifically to the 3' untranslated region of CHIKV viral genome. How does host immune system recognize RNA signatures, and what kind of RNA signatures could be recognized by RNA sensors, are two important questions regarding innate immunity and development of antiviral strategy. This study has explored a new approach to identify immunostimulatory viral RNA signatures and found new features of RLR-mediated RNA sensing. Generally, this study is of significance and broad interests in this field. However, key concerns should be further clarified, and more information should be provided for evaluation of the significance of the current work.

1) This study is based on a HEK293 cell model after overexpression of tagged RLRs. This cell line is different in expression of the three endogenous RLRs. So could these differences affect the enrichment of RNAs for NGS? Moreover, single cell clones were used to perform the infection and RNA isolation, which may also bias the enriched RNAs. In fact, it is better to use a RLR-deficient cell (cell line) with rescue of respective RLR. I may also encourage the authors to use a typical virus-permissive immune cell model instead of HEK293.

2) Previously, some studies have revealed the RNA features of several RNA viruses using different approaches. Despite the proposals of advantages of current approach by the authors, I am more interested in the similarities and differences identified by these researches for RNA viruses. Did common sequence or structural similarities exist for recognition by individual RLR? Otherwise, did common sequence or structural similarities exist in RNA viruses that could be recognized by RLRs? Additionally, did the authors find particular self-RNA in the enriched RNAs? More bioinformatics analysis data should be performed, and these concerns may need further discussions.

3) To evaluate the stimulatory effects of the enriched RNAs, total pooled RNAs were used to activate the IFNβ reporter. Since RNA signatures in MV and CHIKV have been revealed by the study, it may be better for the authors to use these individual "RNA signatures" to examine the effects in IFNβ induction. Considering that IFNβ expression is rather complicated than transactivation, I suggest the authors to use an immune cell to examine the IFNβ levels after transfection of the identified signature RNAs.

---

## [Author Response]

*Although the work is of interest and significance, both reviewers have raised questions regarding your work. We recommend that further analysis of your data may be required; especially how to distinguish your findings from the other similar researches and what are the new concepts revealed by your study.*

The text was modified to discuss these different points in the Discussion part of the manuscript.

[…]

*3) Further bioinformatic assays of the current data are helpful to understand the commons and differences of RNAs recognized by individual RLRs. At least, further description of the detected RNA profiles and the comparison to previous studies is necessary.*

In the revised manuscript, we performed additional bioinformatic analysis of the NGS data. Our studies concluded that not specific RNA sequences but rather structural elements are more likely to be linked to RLRs functionality. The relevant results are presented and discussed.

*4) For the confirmation of the detected RNA signatures, it can be performed in alternative permissive cells that naturally respond to the viruses using the enriched RNAs.* First, we would like to provide a short introduction about MV and CHIKV permissive cells.

For CHIKV the main virus-permissive cells are dermal fibroblasts, migrating monocytes/macrophages, and endothelial cells (Sourisseau et al. 2007,PMID: 17604450; Solignat et al. 2009, PMID: 19732931; Vanlandingham et al., 2005, PMID: 15891138; Kumar et al. PMID: 23253140). Using our RLR-tagged approach we have found that RIG-I specifically interacts with CHIKV 3’-UTR. CHIKV 3’-UTR contains multiple 21 nucleotide-long repeated sequence elements forming dsRNA structure. The evolutional pressure exerted by mosquito vectors on this region and the length of CHIKV 3’-UTR are directly linked to the viral pathogenesis (Chen et al., 2013, PMID: 24009512; Stapleford et al., 2016, PMID: 26807575). We are currently performing additional studies in various cell types in which 3’-UTR of different CHIKV strains and mutant genomes are used. These studies will be later published elsewhere.

Monocytes and dendritic cells of respiratory tract are the first targets of MV that is transmitted *via* aerosols or droplets. Further, MV enters local lymphatic tissues and during the systemicphase of infection MV spreads by mononuclear cells that carry the virus to various organs (skin, intestine, liver, lung and kidney). So in the late stage of the infection epithelial cells are the main targets of MV(Schneider-Schaulies et al., 2001, PMID: 11899069; Esolen et al., 1993, PMID: 8515132; Osunkoya et al., 1990, PMID: 2271420; Yanagi et al., 2006, PMID: 16963735).

Taking into account that immune cells (macrophages, monocytes and dendritic cells) harbour TLR7/8 that recognise viral single- stranded RNA in endosomes of infected cell and lead to activation of the transcription factor NF-κB and activation of type I IFN production, we used in our validation experiments MV-permissive epithelial cells (A549) naturally lacking TLR 7/8 receptors.

In the revised manuscript, a validation of MV RNA signatures for RLR has been added (Figure 10, subsection “Validation of NGS data using conventional approaches”, second paragraph).

Reviewer 1: My major concern is the NGS analysis data. The significant enriched reads represented in red and non-significant reads in blue from Figure 6 to Figure 8. But the normalized read counts are not many in L region of LGP2-MV in Figure 6. The scale of the y-axis is shorter than MDA5-MV, thus how is the specificity of each region represented in this data?

The low number of read counts corresponding to the L region of MV is biologically well known. MV belongs to the order *Mononegavirales,* in which the relative efficiency of expression of the various viral genes depends on a gradient of transcription. Indeed, MV transcription is initiated from a single promoter at the 3’ end of the genome and at each gene junction the virus RNA-dependent RNA-polymerase either falloff or continue the transcription of the downstream gene (Collins et al., 1980, PMID: 7420539; He et al., 1999, PMID: 10400712; Plumet et al., 2005, PMID: 15890929). Thus, due to this gradient of transcription, promoter-proximal genes are expressed more efficiently than promoter-distal genes, a differential level that corresponds to the virus needs. To provide a clear vision of MV transcriptome, we now introduced a figure with NGS analysis of total RNA from MV-infected cells (Figure 6—figure supplement 1) and discussed this in the revised manuscript (subsection “NGS analysis reveals that distinct regions of MV genome are specifically recognized by distinct RLRs”, first paragraph).

The specificity of NGS data analysis refers to the ability of the analysis to recognize truly not enriched positions, avoiding the risk to have false positives. It is impossible to independently assess the specificity of each region (because all positions are considered simultaneously in the analysis), nor to compute this specificity because we do not know which positions are truly not enriched (no reference population). However, we used statistical methods that are well adapted to analyse count data with a low number of replicates. These methods have been shown to be among the most reliable (Soneson and Delorenzi, 2013, PMID: 23497356) and to control the false positive rates to expected levels.

It is true that read counts are low in the L region, but this is the situation where the statistical analysis has less power. So the fact that read counts in this region are significant over the whole L region gives even more confidence about the significance of the difference.

*In Figure 7, where is the position of V region of MVΔV?*

We modified MV genome scheme to show that V region was deleted. We thank the reviewer #1 for this remark (Figure 7).

*In Figure 8, there are many enrichment reads around C/E2 region in RIG-I-CHIKV sample. What is the biological meaning of these differences?* This global enrichment in sequencing reads corresponding to the C-E2-E1 region has also its biological reason. Indeed, previous studies have shown that alphaviruses subgenomic RNA is more abundant than genomic RNA in cytoplasm of infected cells (Pushko et al., 1997, PMID: 9434729; Strauss and Strauss, 1994, PMID: 7968923)

To provide a better vision on CHIKV NGS results, Figure 9 has been modified with a schematic representation of the genomic RNA and the position of subgenomic RNA transcription start site added to the Figure. Also, a new figure (Figure 9—figure supplement 1) was added to present the CHIKV transcriptome (total RNA) and discussed in the revised manuscript (subsection “NGS analysis reveals that RIG-I recognizes specifically the AU-rich 3’ region of CHIKV genome”, second paragraph).

Reviewer 2: 1) This study is based on a HEK293 cell model after overexpression of tagged RLRs. This cell line is different in expression of the three endogenous RLRs. So could these differences affect the enrichment of RNAs for NGS? Moreover, single cell clones were used to perform the infection and RNA isolation, which may also bias the enriched RNAs. In fact, it is better to use a RLR-deficient cell (cell line) with rescue of respective RLR. I may also encourage the authors to use a typical virus-permissive immune cell model instead of HEK293.

We chose HEK293 cells to over-express the tagged RLRs because these cells have a low level of endogenous RLR compared to other cell lines (http://www.proteinatlas.org/). Indeed, we did not detect any of the three RLRs by western blot in the background HEK293 cells when using RLR specific antibodies (Figure 1). RLR specific signals were only observed in cells over-expressing each of the RLRs. The same results were systematically observed in cells infected with MV and CHIKV (data not shown). Additionally, our qPCR analysis of RLR mRNA expression in ST-RLR cells showed that the numbers of mRNA coding for RIG-I and MDA5 were increased 100 fold and for LGP2 10,000 fold in the corresponding ST- RLR cell line (Figure 1). Thus, we believe that we do not need to use “a RLR-deficient cell line with rescue of respective RLR” and that the bias for NGS due to the presence of endogenous RLRs in our ST-RLR cells should be minimal if any.

*I may also encourage the authors to use a typical virus-permissive immune cell model instead of HEK293.*

We are grateful to the reviewer #2 for suggesting this experiment. In the revised manuscript we applied immuno-precipitation of MDA5 from human monocytes infected with MV and validated MDA5 predisposition to bind the MV-N mRNA (Figure 10—figure supplement 1). These results strongly support the specificity of our method.

*2) Previously, some studies have revealed the RNA features of several RNA viruses using different approaches. Despite the proposals of advantages of current approach by the authors, I am more interested in the similarities and differences identified by these researches for RNA viruses. Did common sequence or structural similarities exist for recognition by individual RLR? Otherwise, did common sequence or structural similarities exist in RNA viruses that could be recognized by RLRs?*

RLR specific PAMPs were previously detected for two paramyxoviruses (MV and Sendai virus) by immuno-precipitation of RIG-I and MDA5 and NGS of specific RNA ligands (Baum et al., 2010, PMID: 20805493; Runge et al., 2014, PMID: 24743923). This was already discussed in the previous version of the manuscript. In the revised manuscript we provide further information on the similarities and differences of our results and previous data (Discussion).

*Additionally, did the authors find particular self-RNA in the enriched RNAs?*

Unfortunately, in the current experimental setup we used samples multiplexing for NGS that allowed an appropriate analysis of viral -derived RNA but not cellular RNA molecules. Indeed, the mean number of reads aligned to the human genome in our NGS results were 16x10^[6]^ reads per sample, whereas in order to analyse human transcripts we need at least x10^[8]^ reads/per sample. In the future, our protocol will be adapted to identify RLR-specific cellular partners that will include less or no multiplexing upon NGS.

*More bioinformatics analysis data should be performed, and these concerns may need further discussions.*

In the revised manuscript, we performed additional bioinformatic analysis of the NGS data obtained for the three RLRs upon infection with MV. We evaluated AU bias of the RLR sequencing results (Figure 8) as well as predicted the potential to form RNA secondary structures by different parts of the MV genome (Figure 8, subsection “Primary and secondary structures analysis of RLR-specific viral RNA ligands” and Discussion).

The Results, Discussion, Materials and methods sections as well as the Figure legends were modified accordingly.

3) To evaluate the stimulatory effects of the enriched RNAs, total pooled RNAs were used to activate the IFNβ reporter. Since RNA signatures in MV and CHIKV have been revealed by the study, it may be better for the authors to use these individual "RNA signatures" to examine the effects in IFNβ induction. Considering that IFNβ expression is rather complicated than transactivation, I suggest the authors to use an immune cell to examine the IFNβ levels after transfection of the identified signature RNAs.

Immune cells (macrophages, monocytes and dendritic cells) express TLRs capable of recognizing viral RNA (TLRs 3/7 and 8), and secrete type-I IFN upon ligand binding. In order to assess the immunostimulatory activity of RLR-specific RNA fragments synthesised in vitro and containing the RLR specific signatures, human PBMCs were transfected ex vivo. Indeed, we observed an increased secretion of type-I IFN. However, we were unable to discriminate what percentage of the response was TLR-dependent or RLR- dependent. In the presence of TLR7/8 inhibitor, the cytokine response was strongly reduced (Data not shown). Therefore, using immune cells represented a technical bias that makes it complicated to address this question in the allotted period. However, we believe that studying the RLR response in non-immune cells that do not respond through the TLR pathway is highly pertinent. Indeed, RLRs are ubiquitously expressed, and any cell is susceptible of recognizing viral RNA in their cytoplasm. In addition, non-immune cells are classically used as a model to study the RLR response (Plumet et al., 2005, PMID: 15890929; Runge et al. 2014, PMID: 24743923 Schilte et al., 2012, PMID: 20123960). Consequently, validation experiments were done using MV-permissive epithelial cells naturally lacking TLR7/8 (Figure 10).

The Results, Discussion, Materials and methods sections as well as the Figure legends were modified accordingly.